



Atmospheric
Measurement
Techniques

# Continuous atmospheric CO₂, CH₄ and CO measurements at the Observatoire Pérenne de l'Environnement (OPE) station in France from 2011 to 2018

**Sébastien Conil**[1], **Julie Helle**[2], **Laurent Langrene**[1], **Olivier Laurent**[2], **Marc Delmotte**[2], **and Michel Ramonet**[2]

[1]DRD/OPE, Andra, Bure, 55290, France
[2]Laboratoire des Sciences du Climat et de l'Environnement (LSCE/IPSL), UMR CEA-CNRS-UVSQ, Gif-sur-Yvette, France

**Correspondence:** Sébastien Conil (sebastien.conil@andra.fr)

**Abstract.** Located in north-east France, the Observatoire Pérenne de l'Environnement (OPE) station was built during the Integrated Carbon Observation System (ICOS) Demonstration Experiment to monitor the greenhouse gases mole fraction. Its continental rural background setting fills the gaps between oceanic or mountain stations and urban stations within the ICOS network. Continuous measurements of several greenhouse gases using high-precision spectrometers started in 2011 on a tall tower with three sampling inlets at 10, 50 and 120 m above ground level (a.g.l.). Measurement quality is regularly assessed using several complementary approaches based on reference high-pressure cylinders, audits using travelling instruments and sets of travelling cylinders ("cucumber" intercomparison programme). Thanks to the quality assurance strategy recommended by ICOS, measurement uncertainties are within the World Meteorological Organisation compatibility goals for carbon dioxide (CO₂), methane (CH₄) and carbon monoxide (CO). The time series of mixing ratios from 2011 to the end of 2018 are used to analyse trends and diurnal and seasonal cycles. The CO₂ and CH₄ annual growth rates are 2.4 and 8.8 ppb yr⁻¹ respectively for measurements at 120 m a.g.l. over the investigated period. However, no significant trend has been recorded for CO mixing ratios. The afternoon mean residuals (defined as the differences between midday observations and a smooth fitted curve) of these three compounds are significantly stronger during the cold period when inter-species correlations are high, compared to the warm period. The variabilities of residuals show a close link with air mass back-trajectories.

## 1 Introduction

Since the beginning of the industrial era, the atmospheric mole fractions of long-lived greenhouse gases (GHGs) have been rising. Increases in surface emissions, mostly from human activities, are responsible for this atmospheric GHG build-up. For carbon dioxide (CO₂), the largest climate change contributor, only around half of the additional anthropogenic emissions are retained in the atmosphere, with the remaining 50 % being absorbed by the ocean and the land ecosystems (Le Quéré et al., 2018). For methane (CH₄) the last 10 years are characterised by high growth rates at many observation sites, following a period of stable mole fractions from 2000 to 2007 (Nisbet et al., 2019; Turner et al., 2019). Monitoring the amount fractions of these GHGs is of primary importance for the long-term climate monitoring but also for the assessment of surface fluxes. Remote and mountain atmospheric measurements are needed to assess background mole fractions because they are performed far from anthropogenic sources and/or are located in the free troposphere. Such " global-scale " data are of great value for monitoring the global atmospheric GHG build-up and estimating global-scale fluxes. However, they are not designed to capture the regional-scale signals necessary to assess local- to regional-scale fluxes. The specific purpose of the European Integrated Carbon Observation System (ICOS) is to establish and maintain a dense European GHG observation network to monitor long-term changes, assess the carbon cycle, and track carbon and GHG fluxes. Inverse atmospheric methods combining tall tower network measurements and transport models are important tools for assessing surface GHG fluxes

exchanged with the biosphere and oceans, and estimating the anthropogenic emissions (Broquet et al., 2013; Kountouris et al., 2018). They also offer independent ways to improve the bottom-up emissions inventories required by the international agreement under the United Nations Framework Convention on Climate Change (Bergamaschi et al., 2018; Leip et al., 2018; Peters et al., 2017).

ICOS was established as a European strategic research infrastructure which provides the high-precision observations needed to quantify the greenhouse gas balance of Europe and adjacent regions. It is now a widespread infrastructure made up of three integrated networks measuring GHGs in the atmosphere, over the ocean and at the ecosystem level. Each network is coordinated by a thematic centre that performs centralised data processing. One of the key focuses of ICOS is to provide standardised and automated high-precision measurements, which is achieved by using common measurement protocols and standardised instrumentations. In the atmospheric monitoring network, ICOS targets the World Meteorological Organization (WMO) Global Atmosphere Watch (GAW) compatibility goals (WMO, 2018) within its own network as well as with other international networks. During the preparatory phase, from 2008 to 2013, a demonstration network and new stations were set up with harmonised specifications (Laurent et al., 2017). The Atmospheric Thematic Centre (ATC) performs several metrological tests on the analysers and provides technical support and training regarding all aspects of the in situ GHG measurements (Yver Kwok et al., 2015). The ATC is also responsible for the near-real-time post processing of the measurements (Hazan et al., 2016).

The OPE station was established between 2010 and 2011, under a close collaboration between the French national radioactive waste management agency (Andra) and the Laboratoire des Sciences du Climat et de l'Environnement (LSCE), as part of the demonstration experiment in accordance with ICOS atmospheric station specifications. It is a continental regional background station contributing to the network by bridging the gap between remote stations like Mace Head (MHD) or Jungfraujoch (JFJ) and urban stations like Saclay or Heidelberg. The potential of ICOS continuous measurements of CO$_2$ dry air mole fraction to improve net ecosystem exchange estimates at the mesoscale across Europe was evaluated in Kadygrov et al. (2015). Pison et al. (2018) addressed the potential of the current ICOS European network for estimating methane emissions at the French national scale.

The main objectives of this paper are to describe the OPE monitoring station and the continuous GHG measurement system, to present its performance characteristics and to draw results from the first 8 years of continuous operations.

## 2 Site description and GHG measurement system

### 2.1 Site location

The OPE atmospheric station (48.5625° N, 5.50575° E WGS84, 395 m a.s.l.) is located on the eastern edge of the Paris Basin in the north-east part of France, western Europe, as shown in Fig. 1. The landscape consists of undulating eroded limestone plateaus dissected by a few SE–NW valleys. The station is on top of the surrounding hills in a rural area with large crop fields, some pastures and forest patches. According to Corine Land Cover 2012, the dominant land cover types in the 25/100 km surrounding area are arable land/crops (39 %/44 %), pastures (14 %/18 %) and forest (44 %/34 %). Based on the GEOFLA database from Institut national de l'information géographique et forestière (IGN), the mean population density within a 25/100 km radius from the station is 26/64 (inhabitants km$^{-2}$). The closest small towns are Delouze with 130 people located 1 km to the south-east and Houdelaincourt with 300 people located 2 km to the south-west. The closest cities are Saint-Dizier (45 000 inhabitants) located 40 km away to the west, Bar-le-Duc (35 000 inhabitants) 30 km to the north-west, Toul (25 000 inhabitants) 30 km to the east and Nancy (450 000 inhabitants) 50 km to the east. With 20 000 cars d$^{-1}$, the major road is located 15 km to the north (RN4). The station includes a 120 m tall tower and two portable and fully equipped modular buildings in a 2 ha fenced area. The station infrastructure was built in 2009 and 2010 and the measurements started in 2011.

The OPE station is designed to host a complete set of in situ measurements of meteorological parameters, trace gases (CO$_2$, CH$_4$, N$_2$O, CO, O$_3$, NO$_x$, SO$_2$) and particle parameters (size distribution, absorption and diffusion coefficients, number and mass, chemical composition, radioactivity). The station is part of the French aerosol in situ network contributing to the ACTRIS and AERONET programmes. It is part of the IRSN (Institut de Radioprotection et de Sûreté Nucléaire) network for ambient air radioactivity monitoring. The station also contributes to the French air quality monitoring network and to the European Monitoring and Evaluation Programme.

### 2.2 Local meteorology and air mass trajectories

Local meteorology is monitored using three sets of meteorological sensors located at the three measurement levels on the tower (10, 50 and 120 m a.g.l.). Standard meteorological parameters, temperature, relative humidity, pressure and wind speed and direction, are monitored in compliance with ICOS Atmospheric Station specifications. Minute-averaged data are logged and used to produce hourly mean fields. In addition there is a ground-based weather station operated by Meteo France, the French national weather service providing hourly mean data in compliance with World Meteorological Organization specifications.

(a)                                        (b)

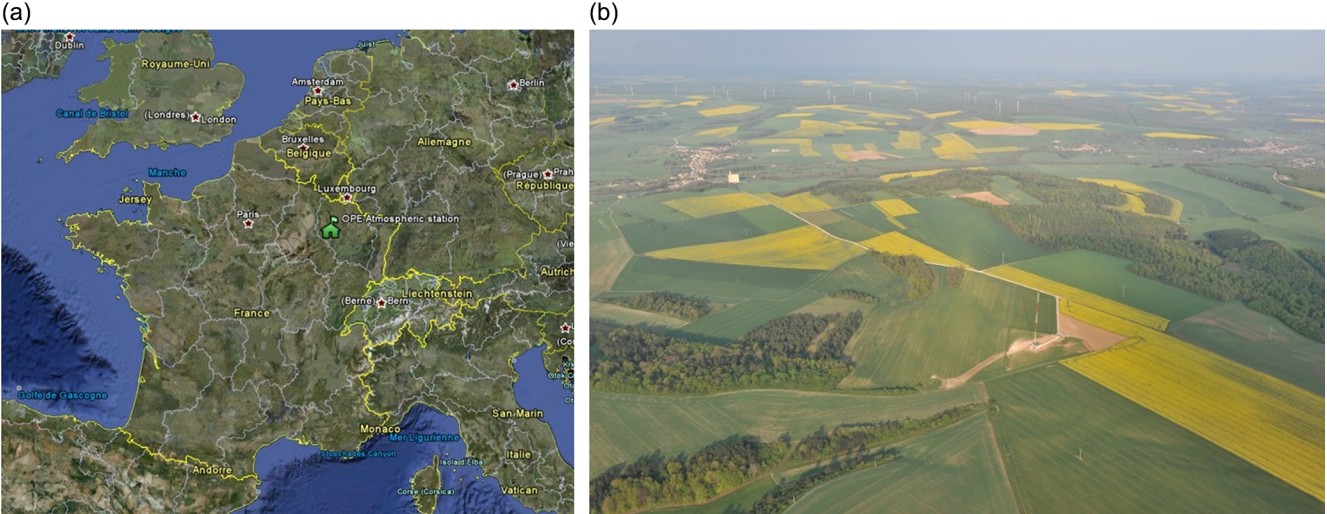

**Figure 1.** Geographical location of the OPE atmospheric station (**a**, ©Google) and aerial photograph illustrating the landscape surrounding the station (**b**).

The mean annual temperature between 2011 and 2018 was 10.5 °C. The minimum temperature was −15.2 °C and the maximum temperature was 36.4 °C. The cumulated annual precipitation was 829 mm on average. Two local wind regimes are predominant, a south-westerly regime and an east-north-easterly regime.

The 96 h back-trajectories were computed for the OPE station top level (120 m) using the National Centers for Environmental Prediction (NCEP) reanalysis fields and the HYSPLIT model every 6 h. As we focus on the afternoon mean residuals (defined as the differences between midday observations and a smooth fitted curve), we only use back-trajectories reaching the OPE station at 12:00 UTC. The clustering tools from HYSPLIT were used to determine the main types of air mass reaching the station. Based on the total spatial variance (TSV) metric, describing the sum of the within-cluster variance, the optimal number of clusters was six (lowest number with a small TSV). The TSV plot is shown in Fig. S1 of the Supplement. The six clusters were defined as shown in Fig. 2. This figure shows the frequency of trajectories for each cluster passing through the corresponding grid point and reaching the OPE station at 12:00 UTC. Clusters 1, 2 and 3 are characterised by continental air masses (mostly from the south, east and north respectively). Cluster 4 is dominated by slow-moving trajectories from the west. Clusters 5 and 6 are dominated by western marine trajectories.

## 2.3 GHG measurement system

The GHG measurement system was set up in 2011 with support from the ICOS Preparatory Phase projects. It was built in order to comply with the Atmospheric Station class 1 station specifications from ICOS. It relies on a fully automated sample distribution system with remote control backed up by

an independent robust spare distribution system. It includes several continuous analysers for the main GHGs (CO$_2$, CH$_4$ and N$_2$O), a manual flask sampler, and specific analysers or samplers for tracers such as radon, CO and $^{14}$CO$_2$.

The continuous GHG measurement system is made of three main parts: an ambient air sample preparation and distribution component, a reference gas distribution component and a master component, which conducts the main analysis sequence and controls the distribution and analysis systems via pressure and flow rate meters. The station flow diagram is described in Fig. 3. Ambient air is collected on the tower at the 10, 50 and 120 m levels and brought down to the shelter located at the tower base using 0.5 in. outer diameter Dekabon tubes fitted with a stainless-steel inlet designed to keep out precipitation. Five sampling lines are installed at 120 m, and three are installed at 10 and 50 m. From the 120 m level, one line is connected to the $^{14}$CO$_2$ sampler built by Heidelberg University. Another sampling line is used to collect weekly flask samples. The continuous GHG measurements are performed using two independent sampling lines. The last line is a spare line, which can be operated in the event of problems on another line or for temporary additional experiments such as independent audits like those performed in 2011 and 2014. At 10 and 50 m, two lines are used for the continuous GHG measurement system. Both of these levels also have a spare line.

At each level, the air is flushed from the tower using three Neuberger N815KNE flushing pumps (15 L min$^{-1}$ nominal flow rate) and cleaned by two 40 and 7 μm Swagelok stainless-steel filters. From each sampling line, a secondary KNF N86KTE-K pump (5.5 L min$^{-1}$ nominal flow rate) is used to sample and pressurise the air (through a 2 μm Swagelok filter) to be dried and then analysed. A flowmeter is used to monitor air flow in the flushing line and a pres-

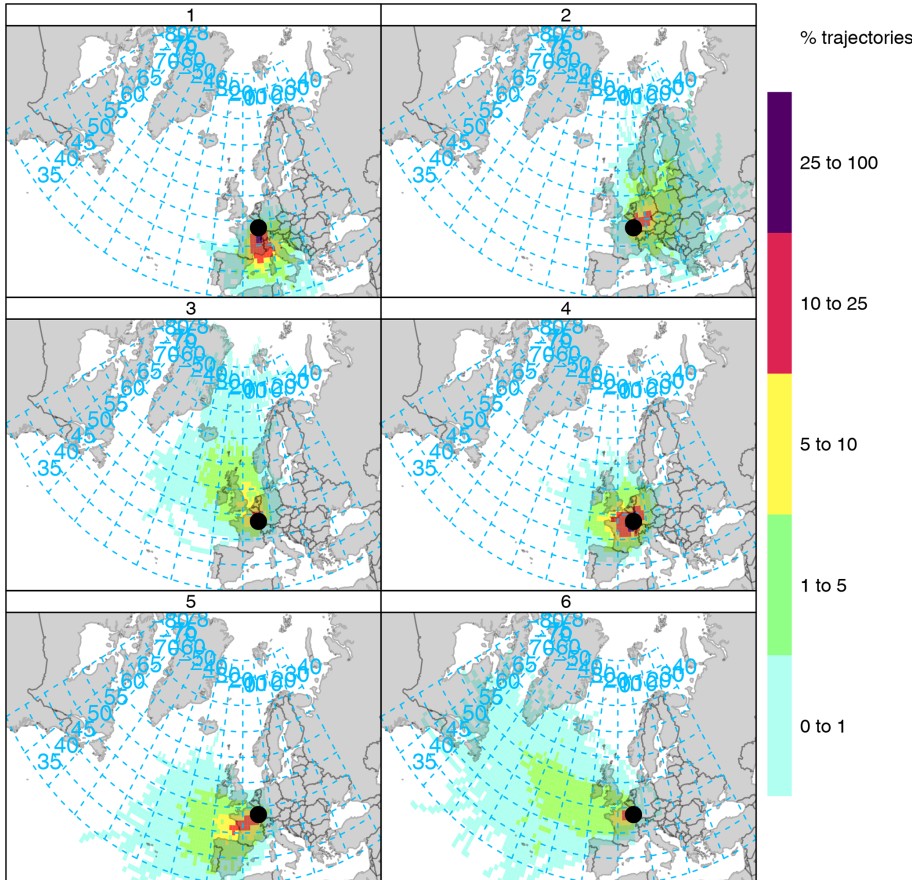

**Figure 2.** The 96 h back-trajectory frequencies reaching the OPE station top level for each of the six clusters identified using the HYSPLIT tools and the NCEP reanalysis for the period 2011–2018.

sure sensor is used to monitor sampling line pressure. The air sample is pre-dried in a coil passing through a fridge. To further dry the sample, the air passes through a 335 mL glass trap cooled in an ethanol bath at $-50\,°C$ using a dewar. Once dried in the cryo-water trap ($-40\,°C$ dew point), the air sample is pressure regulated ($\sim 1150$ hPa absolute pressure at the instrument inlet) and directed to the analysers.

The ambient air distribution component is driven by a control–command component, designed around a programmable logic controller (PLC) for selection and distribution of the ambient air sample from the three sampling heights. This distribution component selects an ambient air sample from one of the three levels using three three-way solenoid valves and then directs it to the drying system and to the analysers. Once analysed, the air sample flows back to the distribution panel where a back pressure regulator controls the air pressure in the sample line. A pressure sensor monitors the pressure at the analyser inlets and a flowmeter monitors the flow rate at the analyser outlets.

The control–command component system selects between standards and ambient air, following the PLC's order, as it is responsible for the sequence management and quality con-

trol processes. The standard gas distribution component is based on a 16-position Vici Valco valve from which nine ports are connected to the analysers. The pressure of the selected standard gas or the ambient air sample is adjusted at the analyser inlet by a manual pressure regulator. All the 1/8 in. or 1/4 in. stainless-steel distributing tubings are over-pressurised to avoid any leakage artefact. According to ICOS internal rules, comprehensive leak checks are performed on a yearly basis and after all maintenance operations.

The analysers used are Picarro series G1000 and G2000 cavity ring-down spectrometers (CRDSs) for $CO_2$, $CH_4$, $H_2O$ and CO and Los Gatos Research off-axis integrated cavity output spectrometers for CO. Each analyser used at the station first underwent extensive laboratory tests at LSCE during the development of the ICOS metrology laboratory at ATC (Lebegue et al., 2016; Yver Kwok et al., 2015). These initial tests provide valuable information about the intrinsic properties of the analysers, their precision, stability, water vapour sensitivity and temperature dependence.

Over the 2011–2018 period, the reference analysers were a Picarro G1301 (ICOS no. 91), which performs $CO_2$ and $CH_4$ (and $H_2O$) mole fraction analyses, and a Los Gatos Research

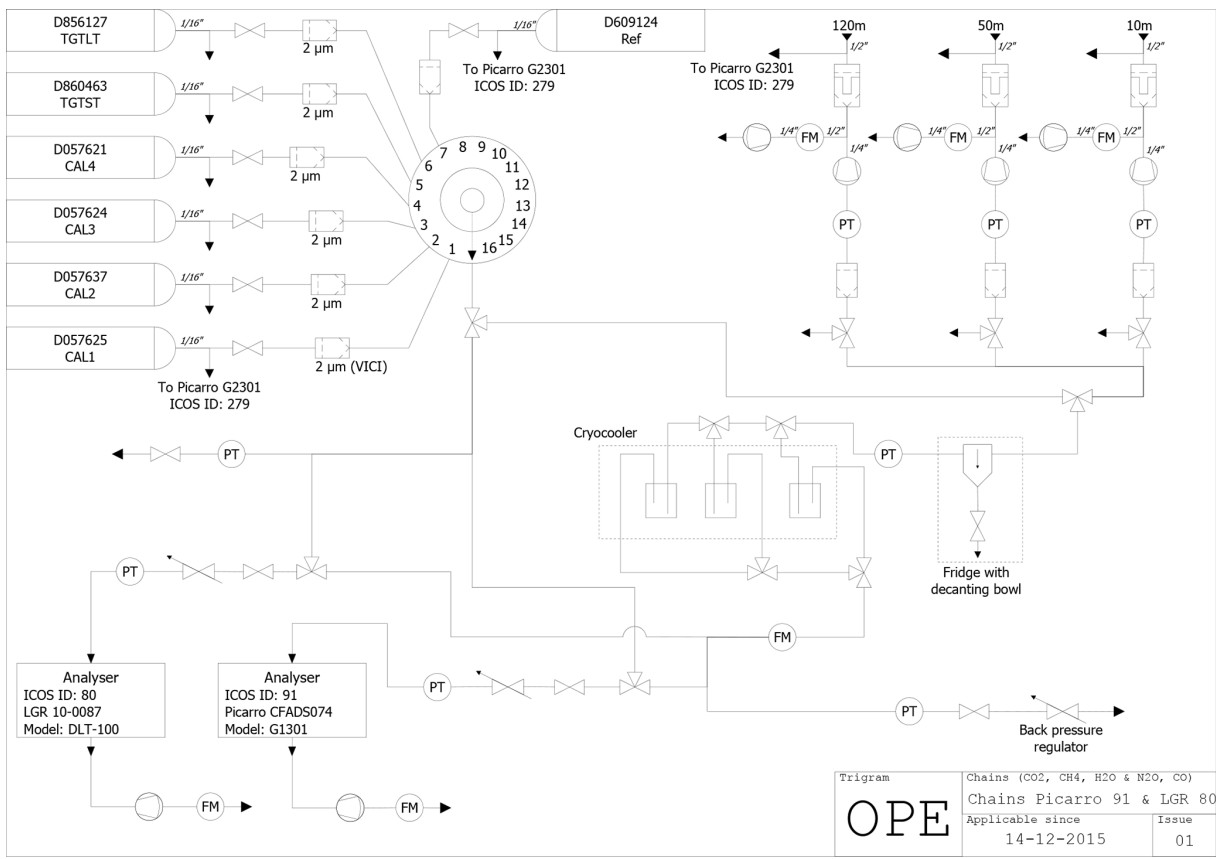

**Figure 3.** Flow diagram of the OPE GHG measurement system (FM: flowmeter; PT: pressure transducer).

DLT100 (ICOS no. 80), which is used for CO (and H$_2$O) mole fraction measurements. A redundant pair of parallel instruments has been running either on the main distribution system or on the spare distribution system using the same calibration and quality control strategy.

The routine operating sequence is as follows:

– a full calibration including four cycles of four standards lasting 8 h followed by 30 min of long-term target (LTT) and then by 30 min of short-term target (STT);

– 5 h of ambient air in cycles of three steps of 20 min for the 10 m level, 50 m level and then 120 m level;

– 20 min of reference gas (REF);

– 5 h of ambient air in cycles of three steps of 20 min of the 10 m level, 50 m level and then 120 m level;

– 20 min of STT.

During the first years of the ICOS preparatory phase, the calibrations were performed every 2 weeks. Due to gas consumption issues and following optimisation tests, the calibrations are now performed every 3 weeks.

The routine sequence is summarised in Table S1 in the Supplement.

The flushing and stabilisation periods for the standards are 10 min, meaning that the first 10 min of data for each of the standards are rejected. The flushing and stabilisation period for the ambient air samples is 5 min, meaning that the first 5 min of data for each of the ambient air levels are rejected (only 15 min of the total 20 min every hour are available). The raw data are then calibrated using the 2- or 3-weekly full calibration and reference working standards following Hazan et al. (2016). Raw data (between 1 and 5 s resolution) are aggregated to 1 min and 1 h averages. The results presented here are based on validated minute data from mid-2011 to the end of 2018.

The calibration strategy includes four consecutive cycles of the four calibration cylinders sampled for 30 min each, the full calibration lasts 8 h. An archive reference standard gas called the long-term target (LTT) is injected every 2 or 3 weeks for 30 min while a common archive reference standard gas called the short-term target (STT) is injected for 20 min every 10 h. Another short-term working standard called the reference (REF) gas is also used every 10 h to correct short-term variability. The mole fractions of the standard cylinders cover the unpolluted atmospheric range following ICOS Atmospheric Station specifications (Laurent, 2017). The standard gases are supplied via SCOTT nickel-plated brass regu-

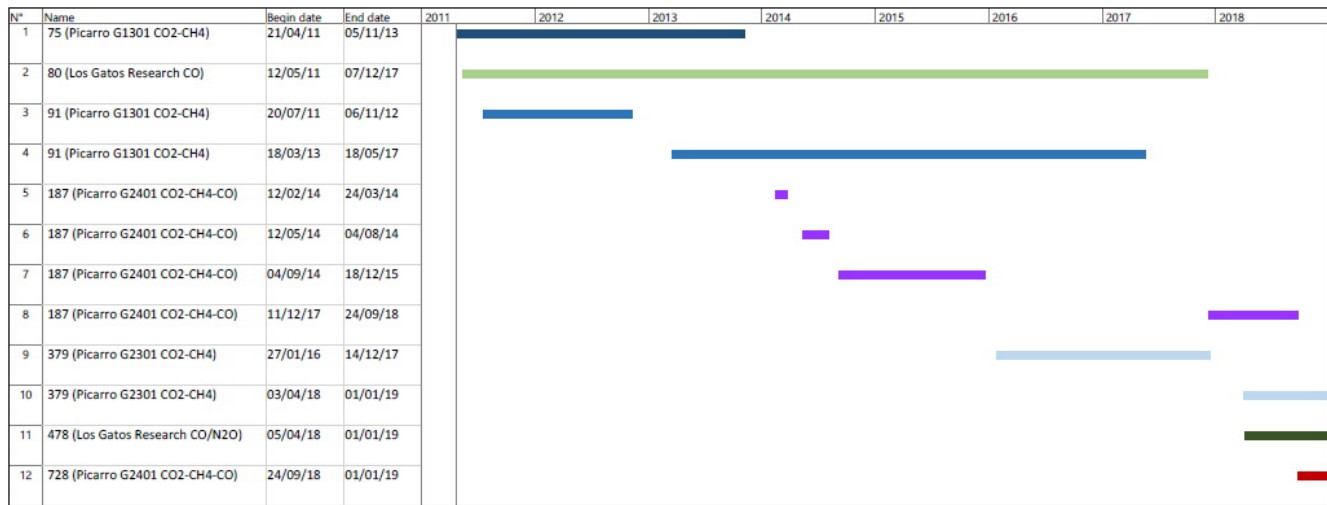

**Figure 4.** Time diagram showing the different GHG analysers in operation at the OPE station.

lators from 50 L Luxfer aluminium cylinders. Before March 2016, the standard and performance cylinders used were prepared by LSCE and were traceable to WMO scales ($CO_2$: WMO X2007; $CH_4$: WMO X2004A; CO: WMO X2014A; $N_2O$: WMO X2006A). Since March 2016, the standard and performance cylinders used have been prepared by the Central Analytical Laboratories of ICOS (CAL) and are traceable to the same WMO scales. STT and REF cylinders are refilled every 6 months by ICOS CAL. All the measurement data presented here were calibrated on these scales.

The raw data from the analysers along with the distribution system monitoring parameters are transmitted to the ATC database on a daily basis. Data are then processed following Hazan et al. (2016) including a specific water vapour correction for the remaining humidity, as well as a station-specific automatic flagging process. Data products are then generated and data quality control is carried out on a regular basis. Additionally manual flagging is performed by the station's principal investigator (PI) on the raw data and on the hourly aggregated data.

Figure 4 gives an overview of the different GHG continuous analysers in operation at the OPE station and their respective time periods. Details on the start and end dates and additional information regarding ancillary instrumentation are given in Table S2 in the Supplement.

## 2.4 Data processing

The GHG data cover several years and were collected using different sampling systems and analysers. In each of the individual time series, some data are missing because of sampling issues, analyser problems or local contamination near the station. Very local pollution, for example due to field works or infrastructure maintenance occurs only rarely. Power outages also occurred due to lighting or construction

work. Problems on the sampling systems are more frequent and include tube leaks, pump troubles, filter clogging or control–command component system failure. Analyser problems are also quite common and range from software issues to operating system failures to hardware problems (hard disk, fan, etc.), or worse, liquid contamination (from water or ethanol) of the optical cell.

Raw data from the instruments (mole fractions and internal parameters such as cell temperature and pressure, outlet valve) and from the air distribution system (sequence information and ancillary data such as pressure and flow rates in the sampling lines) are transferred at least once a day to the ATC data server. Data are then processed automatically as described in Hazan et al. (2016). Sequence data are used to generate ambient air and cylinders' raw time series. Mole fraction raw data are flagged automatically using the ancillary data based on a set of parameters defined for each station and instrument. For the Picarro G1301 no. 91, G2301 no. 379 and G2401 no. 728 analysers, the internal flagging parameters are the same as the ones shown on Table 4 in Hazan et al. (2016). A manual flag is then applied by the station PI in order to eventually discard data using local station information (e.g. local contamination, maintenance operation, leakage, instrumental malfunctions). The list of descriptive flags available to the PI for valid or invalid data is shown in Table 2 of Hazan et al. (2016). Table 1 below presents the quantitative statistical summary of the raw data status for the different instruments used at the OPE station. Details of the internal flagging associated with the flags presented in this table can be found in Table 6 of Hazan et al. (2016). Flag N corresponds to invalid data rejected automatically. Flags O and K correspond to valid and invalid data respectively from the manual quality control. Between 62 % and 72 % of the raw data are valid (O) while around 25 % of the raw data are automatically rejected (N), 20 % being rejected because

**Table 1.** Flags attributed to raw data from the different instruments between mid-2011 and the end of 2018. The last two columns provide the type of flag and the percentage of raw data that were attributed this flag. Flagged O data are valid data manually checked, while N and K flagged are non-valid data automatically and manually rejected respectively.

| Instrument | Compounds | Start | End | Flag | % raw data |
|---|---|---|---|---|---|
| 75 | CO$_2$, CH$_4$ | 21 Apr 2011 | 5 Nov 2013 | O | 72.1 % |
|  |  |  |  | N | 25.8 % |
|  |  |  |  | K | 2.1 % |
| 80 | CO | 12 May 2011 | 7 Dec 2017 | O | 71.0 % |
|  |  |  |  | N | 23.5 % |
|  |  |  |  | K | 5.5 % |
| 91 | CO$_2$, CH$_4$ | 21 Jul 2011 | 22 Jun 2017 | O | 67.2 % |
|  |  |  |  | N | 23.8 % |
|  |  |  |  | K | 9.0 % |
| 187 | CO$_2$, CH$_4$, CO | 12 Feb 2014 | 3 Apr 2018 | O | 65.1 % |
|  |  |  |  | N | 30.7 % |
|  |  |  |  | K | 4.2 % |
| 379 | CO$_2$, CH$_4$ | 27 Jan 2016 | 31 Dec 2018 | O | 71.7 % |
|  |  |  |  | N | 24.9 % |
|  |  |  |  | K | 3.4 % |
| 478 | CO | 27 Jan 2016 | 31 Dec 2018 | O | 62.4 % |
|  |  |  |  | N | 24.9 % |
|  |  |  |  | K | 12.7 % |
| 728 | CO$_2$, CH$_4$, CO | 27 Jan 2016 | 31 Dec 2018 | O | 65.6 % |
|  |  |  |  | N | 25.0 % |
|  |  |  |  | K | 9.4 % |

of stabilisation/flushing. Corrections related to water vapour content and calibration are then applied. Finally, data are aggregated in time to produce minute, hourly and daily means.

From these individual time series, we built three combined time series for CO$_2$, CH$_4$ and CO filling the gaps when possible. The objective is to provide users with continuous time series, combining valid measurements in order to minimise the data gaps. Before merging the time series, each instrument is quality controlled individually, and only measurements which are validated by the automatic data processing and the PI are considered for the combined dataset. For each measurement we indicate the reference of the measuring instrument (unique identifier in the ICOS database), providing the user with analyser traceability. To build these time series from various analyser datasets we used the priority order given in Table 2 for CO$_2$ and CH$_4$ and Table 3 for CO. The priority order is defined a priori by the station PI considering which analysers are fully dedicated to the station for long-term monitoring purposes. In general secondary instruments are installed for shorter periods to perform specific additional experiments (like dry vs. humid air samples, line tests, flushing flow rate tests, etc). For example, 91 was the main instrument for CO$_2$ and CH$_4$ followed by 379. While 91 was in maintenance, instruments 75 or 187 were used as spare instruments. At the beginning of 379 operation, 91 was

still the main instrument, to maintain time series consistency as long as possible. When 91 operation stopped, 379 became the main instrument. When 379 was in repair, instrument 187 was used as a spare instrument again. For CO, the LGR 80 analyser was the main instrument followed by Picarro G2401 728. When the LGR 80 was out of order, we used either Picarro 187 or LGR 478 as spare instruments. When two instruments are installed for long-term measurements, the priority order should take into consideration the performance of each one. It is the responsibility of the station manager to change the priority list in the ICOS database if needed. Merging the individual time series in such a way implies that the merged time series show steps in their uncertainties as individual analysers have different performances (see Sect. 3 for details about the steps in repeatability performance).

Various instruments were used in parallel for some time and it is thus possible to assess systematic differences between the data for these common periods. The instruments may have shared sampling tubes, calibration and quality control gases but may have also used a different air distribution system and different cylinders. Consequently, differences may occur due to problems associated with time synchronisation, air sampling (sampling and flushing pump efficiencies), calibration and water correction or other causes not yet identified.

**Table 2.** Order of priority (main vs. spare analysers) for $CO_2$ and $CH_4$ with ICOS instrument identifiers and the associated period.

| Compound | Main analyser | Spare analyser | Start date | End date |
|---|---|---|---|---|
| $CO_2$, $CH_4$ | | 75 (Picarro G1301) | 21 Apr 2011, 00:00 | 20 Jul 2011, 23:00 |
| $CO_2$, $CH_4$ | 91 (Picarro G1301) | 75 (Picarro G1301) | 21 Jul 2011, 00:00 | 5 Nov 2013, 23:00 |
| $CO_2$, $CH_4$ | 91 (Picarro G1301) | – | 6 Nov 2013, 00:00 | 11 Feb 2014, 23:00 |
| $CO_2$, $CH_4$ | 91 (Picarro G1301) | 187 (Picarro G2401) | 12 Feb 2014, 00:00 | 27 Jan 2016, 00:00 |
| $CO_2$, $CH_4$ | 91 (Picarro G1301) | 379 (Picarro G2301) | 27 Jan 2016, 00:00 | 22 Jul 2017, 00:00 |
| $CO_2$, $CH_4$ | 379 (Picarro G2301) | – | 22 Jul 2017, 00:00 | 14 Dec 2017, 00:00 |
| $CO_2$, $CH_4$ | | 187 (Picarro G2401) | 14 Dec 2017, 00:00 | 3 Apr 2018, 14:00 |
| $CO_2$, $CH_4$ | 379 (Picarro G2301) | – | 3 Apr 2018, 14:00 | 24 Sep 2018, 14:30 |
| $CO_2$, $CH_4$ | 379 (Picarro G2301) | 728 (Picarro G2401) | 24 Sep 2018, 14:30 | – |

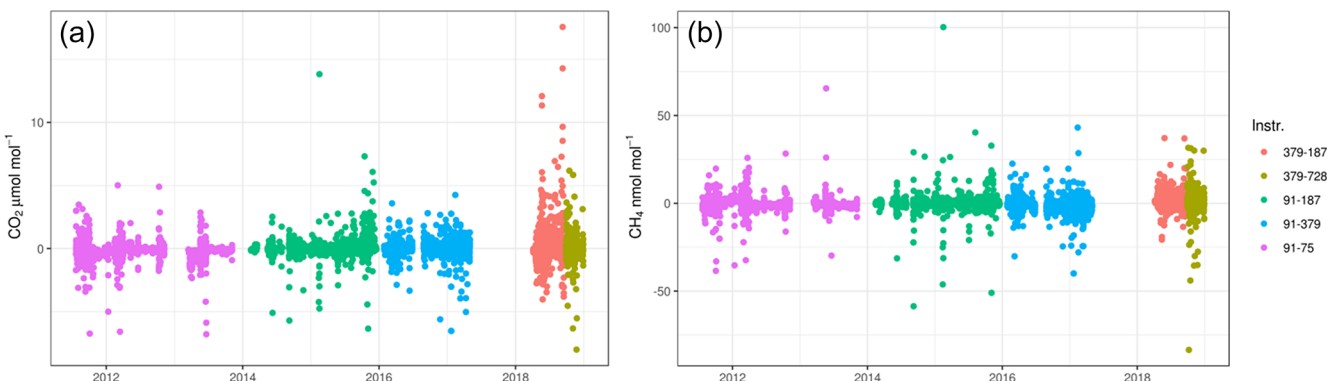

**Figure 5.** Difference between hourly mean afternoon (12:00–17:00 UTC) data at the top level 120 m from the two instruments used at the same time at the OPE station from 2011 to 2018 for $CO_2$ **(a)** and $CH_4$ **(b)**. The different instruments pairs are shown in colour and their identifiers are labelled next to **(b)**.

Figure 5 shows the afternoon (12:00–17:00 UTC) hourly data difference between the different instruments analysing ambient air at 120 m for $CO_2$ and $CH_4$. Large deviations in the afternoon means are revealed by such comparison. Summary statistics for the differences shown in Fig. 5 for the 120 m level (and for the 10 and 50 m levels) are given in Table S3 of the Supplement. On average, over the full period, the differences at 120 m are −0.002 ppm for $CO_2$ and −0.27 ppb for $CH_4$, below the WMO GAW compatibility goals (0.1 ppm for $CO_2$ and 2 ppb for $CH_4$). These significant deviations may come from various sources of uncertainty, such as differing residence time in the sampling systems, water vapour correction, clock issues or internal analyser uncertainties.

No data filtering was applied regarding the differences, and the overall biases are small (Table S3). Large differences can be observed over short periods, especially when the atmospheric signal shows very high variability. For such atmospheric conditions any difference in the time lag between air sampling and measurement in the analyser cell has a significant influence. The persistent presence of a bias between two instruments is used as an indication to perform checks on instruments and air intake chains. For large differences, one of the instruments is generally disqualified based on the tests

performed. In the case of moderate differences, the objective is to use this information for estimating uncertainties.

In a similar approach, Schibig et al. (2015) reported results from the comparison between $CO_2$ measurements from two continuous analysers run in parallel at the JFJ station in Switzerland. The hourly means of the two analysers showed a general good agreement, with mean differences of the order of 0.04 ppm (with a standard deviation of 0.40 ppm). However significant deviations of several parts per million were also found.

## 3  Data quality assessment

QA–QC protocols are applied at several steps in the measurement system. Every day, a conservative quality control is conducted from two complementary standpoints: firstly, intrinsic properties of the spectrometers are verified, and secondly the sampling system parameters are checked. On a weekly to monthly basis, the field performance of the spectrometers is also checked. A flask programme also runs in parallel and is used to expand the atmospheric monitoring to other trace gases and to assess the quality of the continuous measurements. Up to now, flask data were not fully available or were contaminated, and thus have not been used in

**Table 3.** Order of priority (main vs. spare analysers) for CO with ICOS instrument identifiers and associated period.

| Compound | Main analyser | Spare analyser | Start date | End date |
|---|---|---|---|---|
| CO | 80 (Los Gatos CO, N$_2$O) | – | 12 May 2011, 00:00 | 7 Nov 2012, 00:00 |
| CO | 80 (Los Gatos CO, N$_2$O) | – | 11 Mar 2013, 00:00 | 12 Feb 2014, 00:00 |
| CO | 80 (Los Gatos CO, N$_2$O) | 187 (Picarro G2401) | 12 Feb 2014, 00:00 | 18 Dec 2015, 00:00 |
| CO | 80 (Los Gatos CO, N$_2$O) | – | 18 Dec 2015, 00:00 | 7 Dec 2017, 00:00 |
| CO | | 187 (Picarro G2401) | 14 Dec 2017, 00:00 | 5 Apr 2018, 18:00 |
| CO | 187 (Picarro G2401) | 478 (Los Gatos CO, N$_2$O) | 5 Apr 2018, 18:00 | 24 Sep 2018, 14:00 |
| CO | | 478 (Los Gatos CO, N$_2$O) | 24 Sep 2018, 14:00 | 24 Sep 2018, 14:30 |
| CO | 728 (Picarro G2401) | 478 (Los Gatos CO, N$_2$O) | 24 Sep 2018, 14:30 | – |

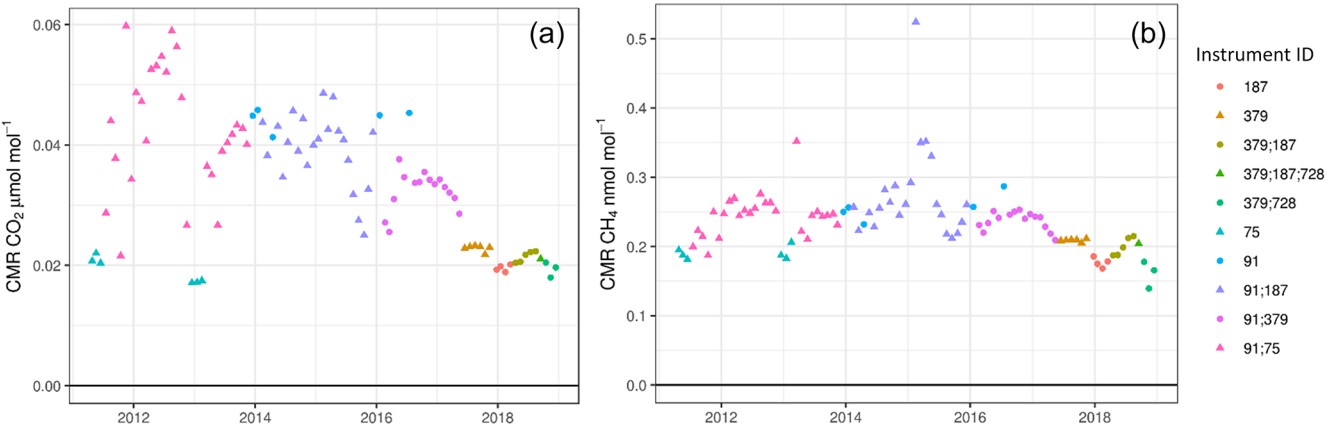

**Figure 6.** Monthly mean continuous measurement repeatability (CMR) field equivalent for CO$_2$ **(a)** and CH$_4$ **(b)** estimated over time for the different instruments in operation at the OPE station over the 2011–2018 period. The different instruments are shown in colour, and their identifiers are labelled in the key by the right panel. Some months have several instruments running at the station and these are identified with several labels.

the present work. A complementary approach to assess compatibility uses round robin or cucumber cylinders circulated between stations within the ICOS European network. Finally, the station compatibility is also assessed during in situ audits using a mobile station and travelling instruments (Hammer et al., 2013; Zellweger et al., 2016).

In this section we used two metrics defined in Yver Kwok et al. (2015) for quality control assessment of the data. These two metrics are usually calculated under measurement repeatability conditions where all conditions stay identical over a short period. Continuous measurement repeatability (CMR), sometimes called precision, is a repeatability measure applied to continuous measurements. Long-term repeatability (LTR), sometimes called reproducibility, is a repeatability measure over an extended period of time. As ICOS targets the WMO GAW compatibility goals within its atmospheric network, the analysers must comply with the performance requirements specified in Table 3 of the ICOS Atmospheric Station specifications report (Laurent, 2017). ICOS precision limits for CO$_2$, CH$_4$ and CO measurements are 50, 1 and 2 ppb respectively. ICOS reproducibility limits for CO$_2$, CH$_4$ and CO measurements are 50, 0.5 and 1 ppb respectively.

## 3.1 Short-term target quality control: continuous measurement repeatability field equivalent

In our basic measurement sequence, the air from a high-pressure cylinder (STT) is analysed twice a day with a 10 h frequency for at least 20 min to assess the daily performance of the spectrometers. This metric mainly describes the intrinsic performance of the spectrometers and not of the sampling system. It is a field estimation of the CMR and is computed as the standard deviations of the raw data over 1 min intervals, the first 10 min of each target gas injection being filtered out as stabilisation.

Figure 6 shows the monthly mean CMR for the combined time series of CO$_2$ and CH$_4$ using the same type of analysers. The time series of CMR for CO are shown in the Supplement (Fig. S2). For CO$_2$, we observe a decrease in the CMR over the measurement periods, indicating an improvement in instrument precision. Analyser no. 91 (Picarro G1301) was shipped to the manufacturer for a major repair including cell replacement between November 2012 and March 2013. The repair at the Picarro workshop improved the CMR performance of the analyser from more than 0.06 ppm to less than 0.05 ppm. For this instrument, the factory estimated a CMR

**Table 4.** Continuous measurement repeatability (CMR) estimated by the factory, MLab and field means over 2011–2018 for $CO_2$ (ppm) and $CH_4$ (ppb). Instrument model and ICOS identifier are indicated in the first columns.

| Analyser | ICOS ID | $CO_2$ (ppm) | | | $CH_4$ (ppb) | | |
| --- | --- | --- | --- | --- | --- | --- | --- |
| | | Factory CMR | ATC Mlab CMR | Field mean CMR | Factory CMR | ATC Mlab CMR | Field mean CMR |
| Picarro G1301 | 91 | 0.04 | 0.059 | 0.048 | 0.27 | 0.24 | 0.27 |
| Picarro G1301 | 75 | 0.019 | 0.022 | 0.02 | 0.18 | 0.26 | 0.22 |
| Picarro G2401 | 187 | 0.023 | 0.026 | 0.021 | 0.2 | 0.28 | 0.22 |
| Picarro G2301 | 379 | 0.025 | 0.023 | 0.022 | 0.23 | 0.22 | 0.2 |
| Picarro G2401 | 728 | 0.014 | 0.013 | 0.014 | 0.1 | 0.09 | 0.08 |

**Table 5.** Continuous measurement repeatability (CMR) and long-term repeatability (LTR)) between factory, MLab and field mean over 2011–2018 of CO (ppb). Their model and ICOS identifier are indicated in the first columns.

| Analyser | ICOS ID | CO (ppb) | | | | |
| --- | --- | --- | --- | --- | --- | --- |
| | | Factory CMR | ATC Mlab CMR | Field mean CMR | ATC Mlab LTR | Field mean LTR |
| Los Gatos $N_2O$ and CO | 80 | 0.15 | 0.06 | 0.06 | 0.3 | 0.4 |
| Picarro G2401 | 187 | 6.5 | 5.7 | 5.17 | 1.7 | 1.18 |
| Los Gatos | 478 | 0.06 | 0.09 | 0.05 | 0.09 | 0.05 |
| Picarro G2401 | 728 | 2.7 | 2.69 | 2.76 | 0.22 | 0.33 |

of 0.04 ppm in 2009, and the lab test at ATC metrology laboratory (MLab) in 2012 estimated a CMR of 0.06 ppm.

Using a gas chromatograph at the Trainou (TRN) tall tower, Schmidt et al. (2014) found a mean standard deviation in the hourly target gas injections of 0.14 ppm for $CO_2$, 3.2 ppb for $CH_4$ and 1.9 ppb for CO for the whole period of 2006–2013. Berhanu et al. (2016) presented the Beromünster tall tower GHG measurement performance using precision, a metric based on the standard deviation of the 1 min target gas measurements, at 0.05 ppm for $CO_2$, 0.29 ppb for $CH_4$ and 2.79 ppb for CO using a Picarro G2401 spectrometer over 19 months from 2013 to 2014. Lopez et al. (2015) presented short-term repeatability (a metric similar to CMR) estimates for the gas chromatograph system used at Puy de Dôme (PDD) at 0.1 ppm for $CO_2$ and 1.2 ppb for $CH_4$, for the years 2010–2013. Table S4 of the Supplement summarises this information.

Table 4 presents the comparison of the $CO_2$ and $CH_4$ CMR for the instruments nos. 75, 91, 187, 379 and 728 estimated by the manufacturer and by the ICOS ATC MLab along with the mean values from station measurements over the 2011–2018 period. The station performance of each individual analyser is consistent with its performance estimated at the factory and at the ATC MLab. Performance is maintained over several years and was not disturbed by the station setting.

For $CH_4$, the factory-estimated CMR for instrument no. 91 in 2009 was 0.27 ppb and the initial lab tests at ATC MLab in 2012 estimated CMR for $CH_4$ to be 0.24 ppb. The repair at

the Picarro workshop did not modify the CMR performance of the analyser. For each instrument, the $CH_4$ performance is very stable over the years with very few outliers.

The CO performance (CMR and LTR) estimated at the station is compared to the factory and ATC MLab results in Table 5.

The CMR time series for CO (Fig. S2 of the Supplement) displays four different periods which are directly linked to the analysers used to build the merged time series. We used two different types of analyser: one built by Los Gatos Research (instruments nos. 80 and 478) and one built by Picarro (instruments nos. 187 and 728). These two types of analyser have very different internal properties as can be seen in Table 5. The CO CMR results reflect such large differences (shown in Fig. S2 of the Supplement), with the CO CMRs from Los Gatos Research instruments being lower than the CO CMRs from Picarro. The Picarro 187 and 728 CO LTRs are significantly lower than their CO CMRs. This means that their raw data have large high-frequency variabilities but when averaged over several minutes these instruments are quite stable (they are not very sensitive to atmospheric or pressure changes).

Overall the precisions measured at the station for $CO_2$, $CH_4$ and CO remain similar to the initial values estimated by the manufacturer and the ATC laboratory, showing no degradation due to the design of the station or the measurement procedures.

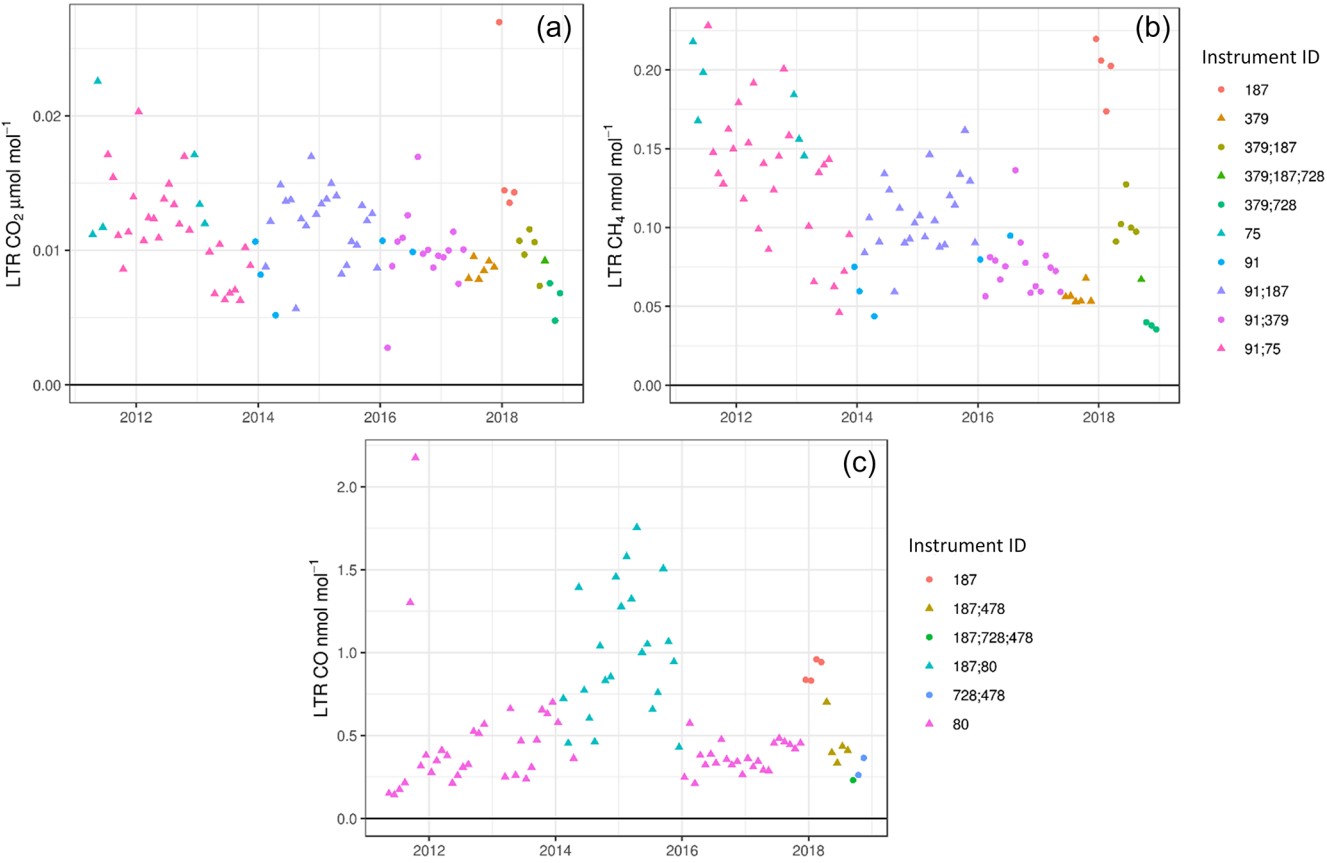

**Figure 7.** Monthly mean field long-term repeatability (LTR) for $CO_2$ **(a)**, $CH_4$ **(b)** and CO **(c)** estimated over time for the different instruments in operation at the OPE station over the 2011–2018 period. The different instruments are shown in colour and their identifiers are labelled in the keys of the top and bottom panels. Some months have several instruments running at the station and these are identified with several labels.

## 3.2 Field long-term repeatability

The field LTR is computed as the standard deviation of the averaged STT measurement intervals over 3 d as performed during the initial test at the ICOS metrology lab. Data are then averaged every month. The same STT data are used but with a different perspective, more closely linked to the ambient air data uncertainty.

Figure 7 shows the monthly mean field LTR of the merged time series using the different instruments and sampling systems. This figure shows the uncertainties of the data related to the analysers (not the sampling systems). As for CMR, $CO_2$ and $CH_4$ LTR show decreasing trends suggesting an improvement of the internal performance of the spectrometers built by Picarro and of the air distribution system and data selection/flagging. The early part of 2018 experienced a markedly worse LTR compared to following months. This is mostly due to the use of instrument no. 187, which has relatively poor performance compared to other instruments.

The comparisons of the field mean LTR and ATC MLab LTR for the different instruments are shown in Table 6 for $CO_2$ and $CH_4$. The LTR field performance of the analysers is consistent with their initial assessments. Periods of lower $CO_2$ and $CH_4$ LTR are associated with instruments no. 91, 379 or 728 while periods with higher $CO_2$ and $CH_4$ LTR are associated with instrument nos. 75 or 187.

As for CMR, the CO LTR monthly time series shows four different periods but with a smaller contrast, associated with the type of analyser used at the station. Most periods with LGR instruments (no. 80 or 478) show a LTR below 0.7 ppb while periods with Picarro instrument no. 187 show a LTR above 0.5 ppb.

Different periods have different uncertainty levels related to instrument performance. While Los Gatos Research instruments show lower CO LTRs they have stronger temperature sensitivities generating high short-term variability in conditions where the temperature is not well controlled. Corrections for these temperature-induced biases required the frequent use of a working standard.

## 3.3 Station audit by travelling instruments

A metric such as CMR is very useful for monitoring the internal performance of instruments and for identifying any

**Table 6.** Long-term repeatability (LTR) of $CO_2$ (ppm) and $CH_4$ (ppb) estimated by MLab and field mean over 2011–2018. Instrument model and ICOS identifier are indicated in the first columns.

| Analyser | ICOS ID | $CO_2$ (ppm) | | $CH_4$ (ppb) | |
|---|---|---|---|---|---|
| | | ATC Mlab LTR | Field mean LTR | ATC Mlab LTR | Field mean LTR |
| Picarro G1301 | 91 | 0.02 | 0.01 | 0.08 | 0.08 |
| Picarro G1301 | 75 | 0.01 | 0.01 | 0.21 | 0.17 |
| Picarro G2401 | 187 | 0.02 | 0.02 | 0.22 | 0.17 |
| Picarro G2301 | 379 | 0.007 | 0.009 | 0.1 | 0.06 |
| Picarro G2401 | 728 | 0.005 | 0.008 | 0.06 | 0.02 |

instrument failure as early as possible. Other instrument-related metrics such as long-term calibration drift or calibration stability over the sequences are also useful for monitoring instrument performance. However, they do not give an assessment of the overall measurement systems. Flask versus in situ comparisons or station audit by travelling instruments are recognised as essential tools in the performance and compatibility assessment of a measurement system. ICOS audits are performed by a mobile lab, hosted by the Finnish Meteorological Institute in Helsinki, and equipped with state-of-the-art GHG analysers and travelling cylinders. The measurement data from the station are centrally processed at the ATC. However, the data produced by the mobile lab are computed separately to maintain the independent nature of the Mobile Lab and at the same time to evaluate the performance of the centralised data processing.

The OPE station was audited twice, once in summer 2011, soon after the station was set up, during the feasibility study for the travelling instrument methodology, and then in summer 2014, when the ICOS mobile lab was ready for operation. During the 2-week intercomparison in 2011, significant differences for $CO_2$ and $CH_4$ were noticed between the Fourier transform infrared (FTIR) travelling instrument and the CRDS reference instrument (Hammer et al., 2013). As the two instruments have different temporal resolutions and different response times, the CRDS measurements were convoluted with an exponential smoothing kernel representing a 3 min turnover time to match the FTIR specifications. For $CO_2$ the smoothed differences vary between 0.1 and 0.2 ppm with a median difference of 0.13 ppm and a scatter of the individual differences of approximately ±0.15 ppm. The smoothed $CH_4$ differences decrease from 0.7 ppb initially to 0.1 ppb, the median difference being 0.4 ppb. Such large differences were caused by relatively poor performance of the CRDS and FTIR instruments because of specific hardware problems and also due to the large temperature variations (10 K) within the measurement container. During the summer of 2011, the travelling instrument was also set up at the Cabauw (CBW) station in the Netherlands. The audit showed better instrument performance but the same kind of differences for ambient air comparisons. While the $CO_2$

deviations at CBW were partly explained by a travelling instrument intake line drawback and by calibration issues on the main measurement system, at OPE no final explanation has been found for the observed differences.

In the summer of 2014, the 2-month audit was performed using a Picarro G2401 travelling instrument and a FTIR. However the FTIR performance was not yet optimised and the difference in time resolution made it difficult to use it properly. Results from this instrument are not considered here. On average, the OPE standard cylinders analysed by the travelling instrument (TI) showed 0.03 and 0.10 ppm higher $CO_2$ mole fractions at the beginning and at the end of the audit respectively than the assigned values used to calibrate measurements at OPE. Similar results were found for $CH_4$ with relatively low differences ranging between 0 and 1 ppb. The instruments and the working standards (OPE and travelling standards) were calibrated against two different sets of standards, introducing biases in the measurements of cylinders and of ambient air. The intercomparison was complicated by the fact that the station was struck by lightning three times during the summer, causing major power outages and electrical damage to the infrastructure. Such power outages generate shifts in the CRDS analyser response that prevent drift correction of the calibration response, degrading analyser performance. The ambient air comparison was based on two sampling lines, one line delivering dry air samples to Picarro G1301 no. 91 and wet air samples to Picarro G2401 no. 187, and one independent line for the audit supplying wet air samples to the TI. The wet air measurement data from analyser no. 187 data were corrected for water vapour by the factory Picarro correction, but the TI wet air measurement data were corrected by an improved water correction based on a water droplet test performed at the beginning of the intercomparison using a simplified version of the EMPA method no. 2 implementation presented in Rella et al. (2013). The ambient air $CO_2$ mole fractions measured in dry and wet air samples by the OPE analysers showed lower mole fractions compared to the TI measurements, by 0.10 ppm at the beginning of the audit and 0.13 ppm at the end. Most of the differences in ambient air measurements can be explained by the bias in the reference scales.

When averaged over the whole period, the OPE minus TI measurement differences remain within the WMO GAW compatibility goal. The OPE Picarro G1301 no. 91 dry air measurements deviated on average by $-0.05$ ppm compared to the travelling Picarro G2401 wet air measurements in the case of CO$_2$, and by 0.70 ppb in the case of CH$_4$. Similarly the OPE Picarro G2401 no. 187 wet air measurements differ from the TI wet air measurements by $-0.03$ ppm and 1.80 ppb for CO$_2$ and CH$_4$ respectively. The CO comparison was carried out for OPE LGR and OPE G2401 instruments and compared to the TI G2401: the average deviations exceeded the WMO GAW component compatibility goal ($\pm 2$ ppb).

Vardag et al. (2014) presented similar intercomparison results at MHD over 2 months in spring 2013. For CO$_2$, the difference between the TI and the station analyser (Picarro G1301) for ambient air measurements at MHD was $0.14 \pm 0.04$ ppm. During this intercomparison there were no calibration issues as the same set of calibration cylinders was used on both systems. However there could also have been a bias in the water correction effect. Still, most of the differences between station data and the TI during ambient air measurements remained unexplained. These results and the previously published results highlight the major difficulties that station PIs are facing with intercomparison interpretation and understanding. Upcoming sampling line tests, which are mandatory in the ICOS network at least on a yearly basis, may help us understand if the sampling design introduces artefacts.

## 3.4 Travelling cucumber cylinders and station target tank biases

At the beginning of station operation, quality control tanks, or targets, were not systematically used or calibrated. Calibrated tanks were used systematically from 2015 as working standards in order to monitor biases.

In addition the OPE station took part in the CarboEurope "Cucumber" programme in the EURO2 loop at the end of 2014, as well as in the ICOS programme, which started in September 2017. The aim of these programmes is to assess measurement compatibility and to quantify potential offsets in calibration scales within a network. The results of these two sequences of cucumbers intercomparison are shown in Fig. 8 along with the biases estimated for the station quality control cylinders.

The biases estimated from the target tanks operated at the station and the blind Cucumber intercomparison biases are consistent for all species. CO$_2$ biases are found to be between $-0.1$ and 0.1 ppm most of the time except for some outliers that still need to be understood. A slight trend may be present in the LTT CO$_2$ biases between 2014 and 2018. The STT results may show a trend as well but step changes are also present. We attribute the CO$_2$ biases signal to the convolution of step changes and an interannual trend. The step changes may be due to cylinder changes. The possible CO$_2$ trend shown by the LTT (of the order of $+0.02$ ppm) remains unexplained at this stage. The re-evaluation of the CO$_2$ mole fractions of calibration tanks at the ICOS central facility could show a drift in their values, which would lead to a correction of the time series.

CH$_4$ biases are between $-0.75$ and 0.75 ppb for most cases. CO biases show a large spread at the beginning of station operation partly related to the temperature sensitivity of the Los Gatos Research analyser and the poor temperature control of the measurement container. Since 2016 the CO biases stay within the $-5/+5$ ppb range.

## 4 Results

Tall tower GHG mole fraction time series over mid-latitude continental areas exhibit strong variations from hours to weeks, seasonal and interannual timescales, and even longer. Such variabilities are linked to local, regional and global meteorological variations, as well as to land biosphere processes and human activities. We will first show the general characteristics of the time series. We will then analyse and show the diurnal cycles computed from the despiked hourly data. We will select only stable situations with low fast variability to focus on the regional scale and compute afternoon means for CO$_2$, CH$_4$ and CO at the three sampling levels. The seasonal cycles and long-term trends will then be analysed and presented.

## 4.1 General characteristics of the CO$_2$, CH$_4$ and CO time series

Figure 9 shows the general characteristics of the afternoon mean mole fractions for CO$_2$, CH$_4$ and CO at the OPE station at 10, 50 and 120 m above ground level.

From the summer of 2011 to the end of 2018, the afternoon mean CO$_2$ at 120 m varied from 375 ppm to a maximum of 455 ppm. Over this 7-year period, the afternoon mean time series show synoptic variations as well seasonal variations and interannual trends. Similar patterns were observed at several other long-term monitoring stations in western Europe over different periods (Popa et al., 2010; Vermeulen et al., 2011; Schmidt et al., 2014; Lopez et al., 2015; Schibig et al., 2015; Satar et al., 2016; Stanley et al., 2018; Yuan et al., 2019). At European background stations such as the MHD coastal station or mountain stations (JFJ, Zugspitze-Schneefernerhaus (ZSF) or PDD) the interannual times series are dominated by long-term trends and seasonal changes. At regional continental stations (CBW, TRN or Białystok, BIK), the synoptic variations have a much larger intensity due to the proximity of strong continental sources. The patterns and amplitude of synoptic variations and of seasonal changes depend on the sampling height, with the lowest level (10 m) having a larger variability than the highest level (120 m). Ver-

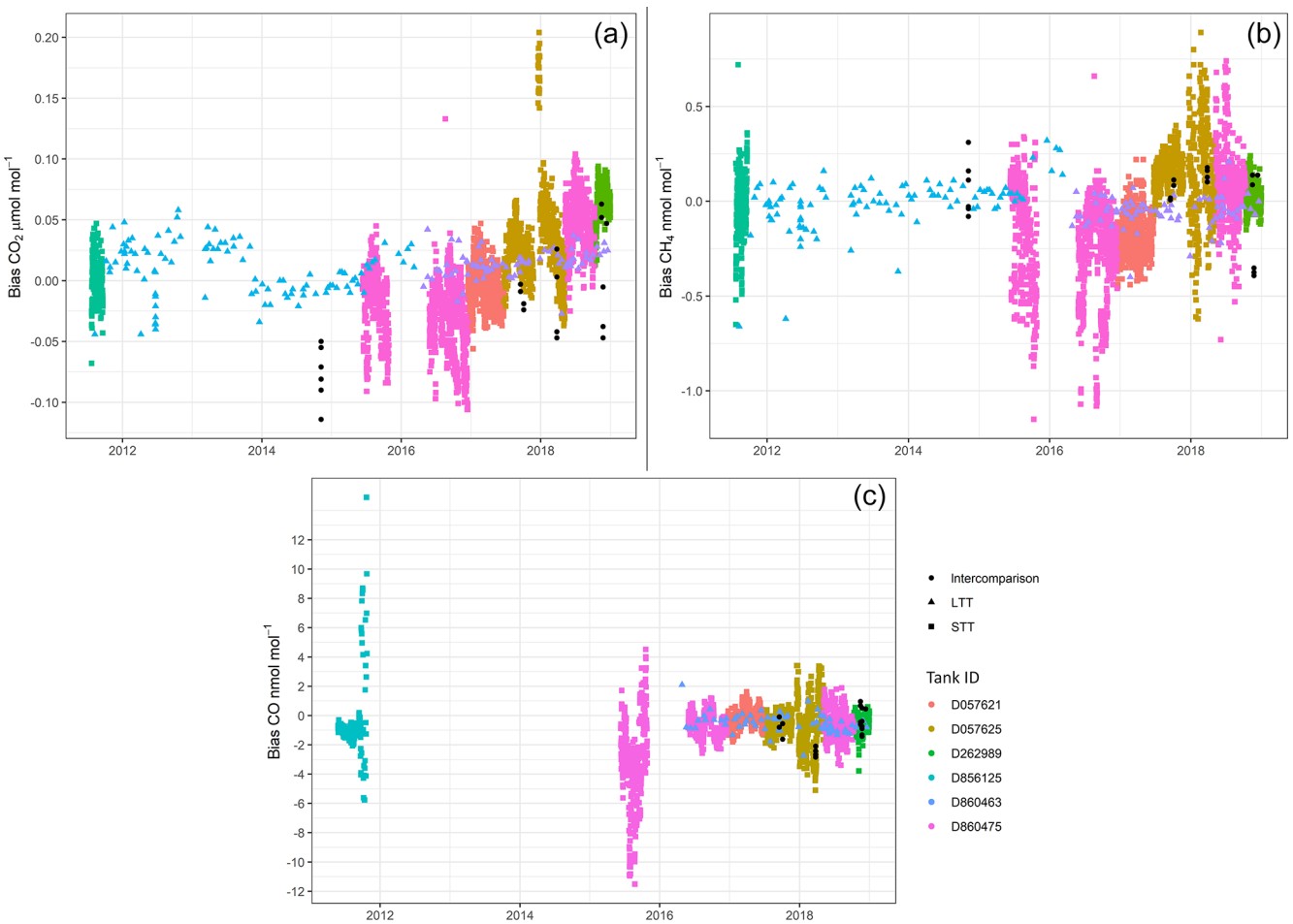

**Figure 8.** Target tank biases over time for several tanks for $CO_2$ **(a)**, $CH_4$ **(b)** and CO **(c)**. The short-term target (STT), long-term target (LTT) and "cucumber" intercomparison biases are shown as coloured squares, coloured triangles and black circles. The different colours are related to the different tanks used at the OPE station for quality control.

tical gradients of $CO_2$ are present year-round but are stronger in summer and weaker in winter, and the gradient variability is much stronger in summer.

The time series for $CH_4$ afternoon mean mole fractions are also characterised by a long-term trend with a weaker seasonal cycle. Synoptic variations can be as high as 150 to 200 ppb on hourly timescales and are stronger at the lowest level. Vertical gradients of $CH_4$ are present year-round and show a small seasonal cycle. The time series for CO afternoon mean mole fractions do not show any long-term trend but are characterised by strong seasonal cycles. Synoptic variations can be as high as 200 ppb on hourly timescales and are stronger at the lowest level. Vertical gradients of CO are much stronger in winter and weaker in summer.

### 4.2 Diurnal cycles and vertical gradients

The diurnal cycles of trace gases result from atmospheric dynamics (especially the daily amplitude of the boundary layer height), surface fluxes and atmospheric chemistry. The mean

diurnal cycles of $CO_2$, $CH_4$ and CO are shown in Fig. 10 for the three sampling levels (10, 50 and 120 m). Despiked hourly data (not detrended or deseasonalised) were used to compute the mean diurnal cycles. $CO_2$, $CH_4$ and CO mole fractions display similar diurnal cycles due to the similar atmospheric dynamics control: a large increase in mean mole fractions and vertical gradient during night-time in contrast to a reduction in the mean of mole fractions and vertical gradients during daytime. During the afternoon, while the $CH_4$ and CO mole fractions at the lowest level stay larger than those at the top level, the $CO_2$ mole fractions at the lowest level are slightly lower than those at the higher level. This $CO_2$ depletion is due to vegetation growth and photosynthesis (which are stronger in summer and almost disappear in winter). The diurnal cycles of $CO_2$ and $CH_4$ are larger in spring and summer while for CO they are larger in winter.

For the three compounds, the vertical gradients are much stronger at night and the highest mole fractions are measured near the ground. During the day, the gradients almost disap-

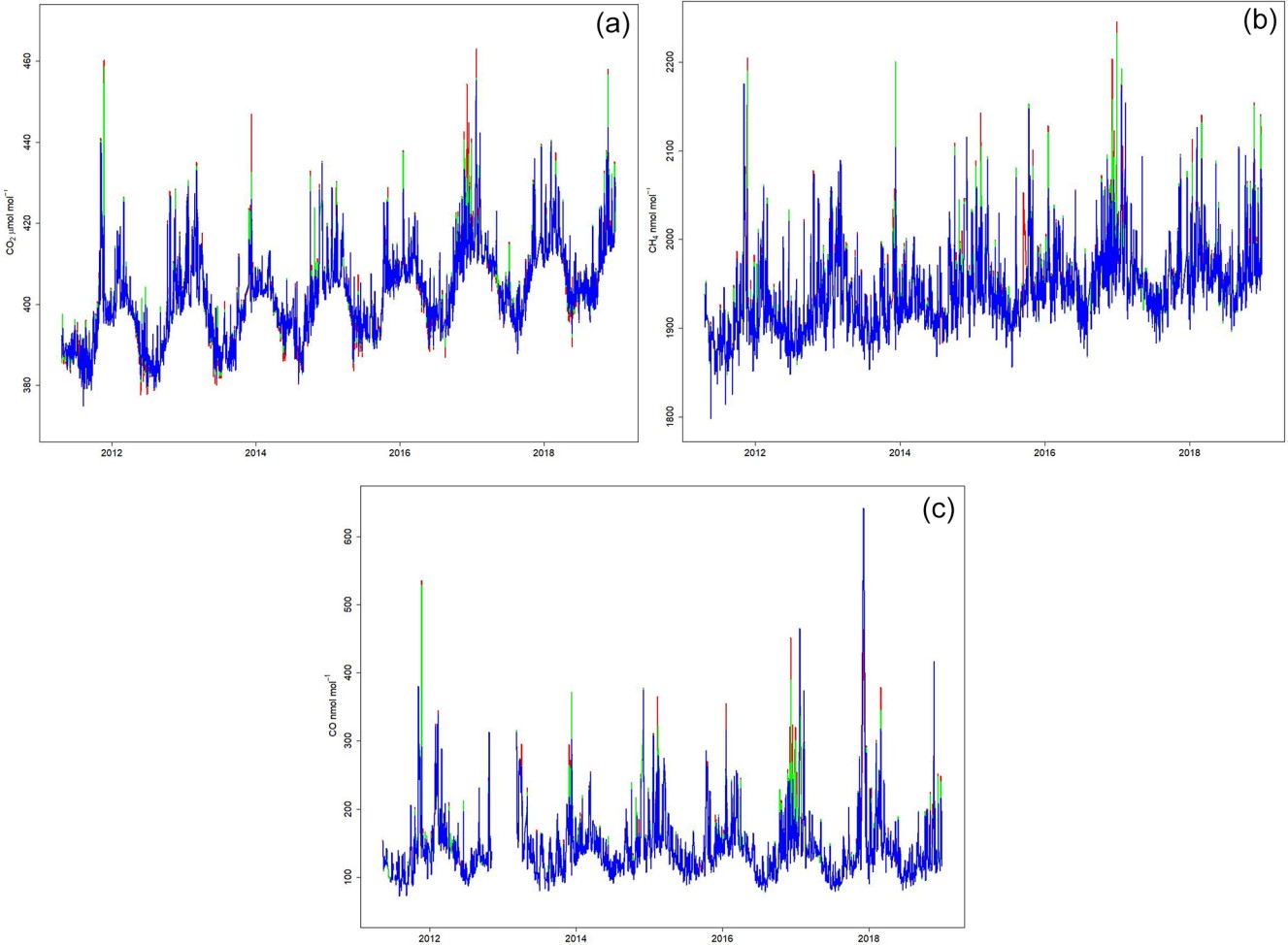

**Figure 9.** Afternoon (12:00–17:00 UTC) mean CO₂ **(a)**, CH₄ **(b)** and CO **(c)** mole fractions measured at the OPE station at 10 m (red), 50 m (green) and 120 m (blue).

pear, mainly because of the enhanced vertical mixing of the lower atmosphere. In spring and summer, the CO₂ afternoon mole fraction at the lowest level is slightly below that at the highest level, reflecting the photosynthesis pumping of CO₂ by plants. Vertical CO₂ gradients build up again in the late afternoon.

In the warm period (from May to September), the mean vertical gradient of CO₂ is 0.4 ppm during the afternoon (12:00–17:00 UTC) and −9.95 ppm at night (00:00–05:00 UTC). During the cold period (from October to April) the mean vertical gradient of CO₂ is −0.24 ppm during the afternoon (12:00–17:00 UTC) and −3.5 ppm at night (00:00–05:00 UTC). Similar patterns were observed at CBW for the 1992–2010 period but with stronger amplitude (Vermeulen et al., 2011). Stanley et al. (2018) showed the vertical gradients of CO₂ and CH₄ mole fractions at two tall towers in the United Kingdom (UK). Daytime vertical differences of CO₂ were very small (< 1 ppm) (positive in winter and neg-

ative in the other seasons). Night-time vertical gradients of CO₂ were always negative between 3 and 8 ppm.

In the warm period the mean CH₄ vertical gradient is −0.5 ppb during the afternoon (12:00–17:00 UTC) and −20.7 ppb at night (00:00–05:00 UTC). In the cold period the mean CH₄ vertical gradient is −4 ppb during the afternoon and −18.5 ppb at night. Similar patterns and amplitudes were shown in the UK by Stanley et al. (2018). Vermeulen et al. (2011) also presented similar patterns but with larger amplitudes, with the CBW vertical gradients of CH₄ reaching −300 ppb during summer between the 20 and 200 m levels.

## 4.3 Regional-scale signal extraction

The station time series exhibit strong variability from hourly to interannual timescales. These variations may be related to meteorological variability and to variations in sources and sinks. We are mostly interested in the regional signatures at scales that can be approached using model inversions and assimilation tools. For this reason, we want to isolate the situa-

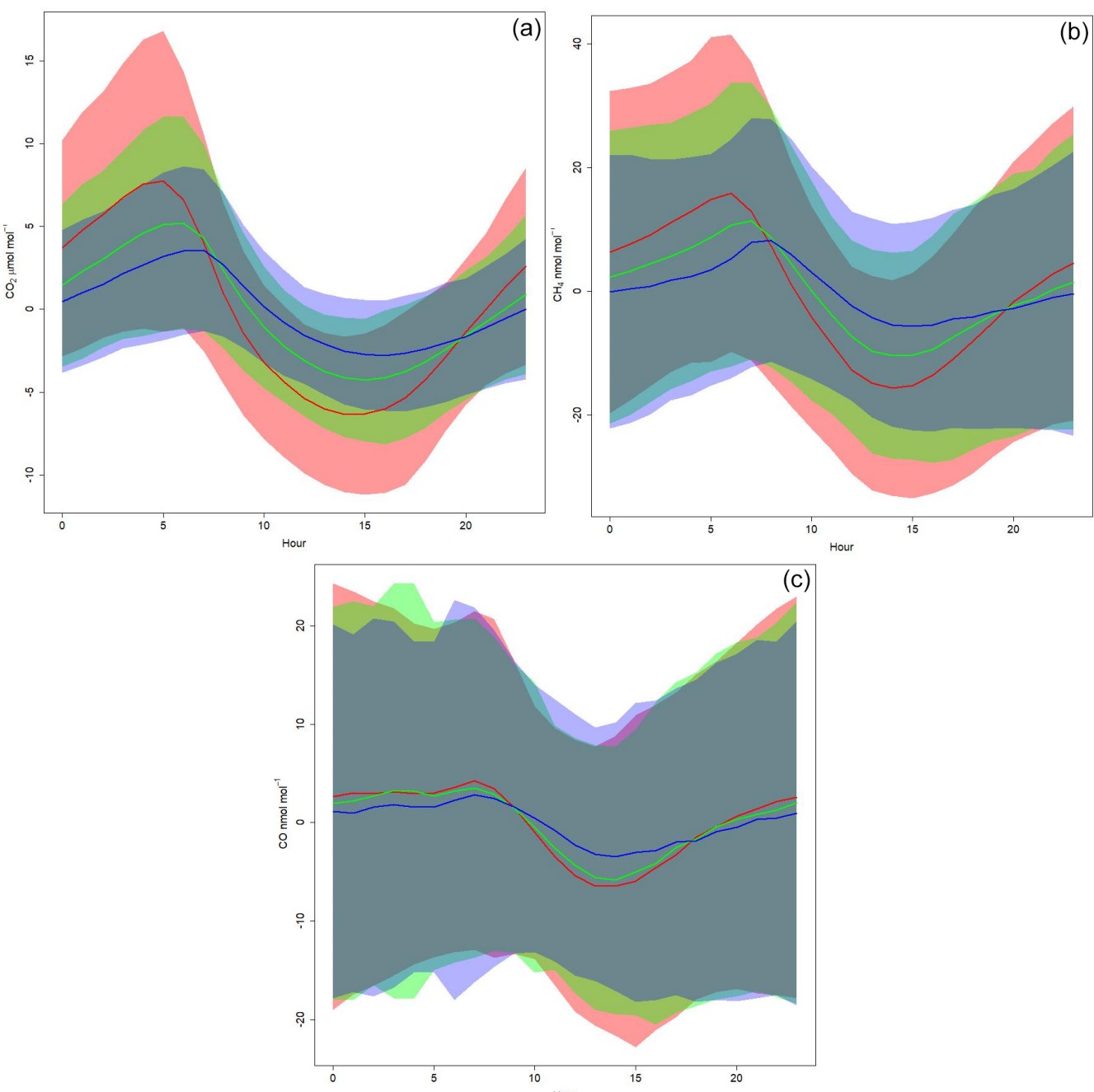

**Figure 10.** Mean diurnal cycles of CO$_2$ **(a)**, CH$_4$ **(b)** and CO **(c)** for the three sampling levels 10 m (red), 50 m (green) and 120 m (blue), computed over the period 2011–2018. The shaded areas correspond to the + and −1 standard deviations around the mean diurnal cycles.

tions where the local influence is dominant and shadows the regional signature from the time series and data aggregation. We then need to define the background signal to which the regional-scale signal is added.

Such local situations and background definitions may be extracted purely from time series analysis procedures, or may be constrained on a physical basis. El Yazidi et al. (2018) assessed the efficiency and robustness of three statistical spike

detection methods for CO$_2$ and CH$_4$ and concluded that the two automatic methods, namely standard deviations (SDs) and robust extraction of baseline signal (REBS), could be used after a proper specification of parameters. We used the El Yazidi et al. (2018) method on the composite merged minute time series to filter out spike situations. From the despiked minute data we built hourly means, which were used to analyse the diurnal cycles. Focusing on data with regional

footprints, we selected only afternoon data with low hourly variability when the boundary layer is larger and the vertical mixing is more efficient. We excluded data showing large variations by using the minute standard deviations. Hourly data with minute standard deviations larger than the three interquartile ranges computed month by month were excluded from the afternoon mean, leading to a rejection of 2.9 % to 4.2 % of the hourly means of $CO_2$, $CH_4$ and CO.

We then used the CCGCRV curve fitting programme from NOAA (Thoning et al., 1989) with the standard parameters set (npoly = 3, nharm = 4) to compute the mean seasonal cycles and trends for the three compounds. CCGCRV results were compared with similar analysis performed using the R package openair (Carslaw and Ropkins, 2012) for the seasonal cycle and the trend using the Theil–Sen method (Sen, 1968). We then computed the afternoon mean residuals from the seasonal cycle and trends using the CCGCRV results.

## 4.4 Seasonal cycles

Figure 11 shows the mean seasonal cycles of $CO_2$, $CH_4$ and CO at the three measurement levels (10, 50 and 120 m a.g.l.). Each of the three GHGs displays a clear seasonal cycle, with higher amplitudes at the lower sampling levels. Minimum values are reached during summer when the boundary layer is higher and the vertical mixing is more efficient. In addition to the boundary layer dynamics, the seasonal cycles of the surface fluxes and of the chemical atmospheric sink also play significant roles. The correlations of dynamic and flux processes at the seasonal scale make it difficult to distinguish the role of each process. $CO_2$ vertical gradients are observed in late autumn to early winter when the $CO_2$ mole fractions at 10 m are larger than at 120 m.

Minimum values are reached in late summer for $CO_2$, around the end of August with no vertical gradients around this minimum. Vertical gradients appear in late spring with a maximum gradient in June when a secondary minimum is observed at the lowest level but not at the higher levels. The amplitude of the $CO_2$ seasonal cycle is nearly 21 ppm at the three levels. The $CO_2$ seasonal cycle amplitudes observed at BIK and CBW were between 25 and 30 ppm depending on sampling height (Popa et al., 2010; Vermeulen et al., 2011). The two early and late summer $CO_2$ minima were also observed by Haszpra et al. (2012) at the Hegyhátsál tall tower in western Hungary between 2006 and 2009, and their timings were very close to those of OPE. But only one summer minimum between August and September was observed at the BIK (Popa et al., 2010), CBW (Vermeulen et al., 2011) and TRN tall towers (Schmidt et al., 2014) and at the Schauinsland (SSL) and ZSF mountain stations (Yuan et al., 2019). Ecosystem $CO_2$ flux measurements performed in 2014 and 2015 near the OPE atmospheric station revealed that the forest and grassland net ecosystem exchange had two maxima in early summer and late summer with a decrease in between (Heid et al., 2018). The two early and late winter

maxima were also observed by Popa et al. (2010) at the BIK tall tower with similar timings, end of November and February. But only one winter maxima was observed in January at CBW (Vermeulen et al., 2011), TRN (Schmidt et al., 2014) and Hegyhátsál (Haszpra et al., 2012), in February at SSL, and in March at the ZSF mountain station (Yuan et al., 2019).

At OPE minimum $CH_4$ values are observed in July and maximum values are reached in February and November. The peak-to-peak amplitude of the $CH_4$ seasonal cycle is nearly 70 ppb at the three levels. At BIK, there was only one maximum in December and minimum values were reached between May and June (Popa et al., 2010). The seasonal cycle amplitude was between 64 and 88 ppb. At CBW, $CH_4$ mole fractions peaked at the end of December and were at a minimum at the end of August. The seasonal cycle amplitude was between 50 and 110 ppb depending on the sampling level (Vermeulen et al., 2011).

The CO seasonal cycle peaks at the end of February, with a secondary peak at the end of November. Minimum values are reached in July, earlier than the $CO_2$ and $CH_4$ minimum. The peak-to-peak amplitude of the CO seasonal cycle is between 80 and 90 ppb. At BIK, the CO maximum was reached in January (with a delay compared to $CO_2$ and $CH_4$) and minimum values were observed in June, with a peak-to-peak seasonal cycle amplitude between 130 and 200 ppb (Popa et al., 2010). At CBW, the CO maximum was reached in January (also with a delay compared to $CO_2$ and $CH_4$) and minimum values were observed in August. The peak-to-peak CO seasonal cycle amplitude varied between 90 and 130 ppb (Vermeulen et al., 2011).

## 4.5 Trends

Table 7 reports the mean atmospheric growth rates computed for the three compounds at the top level using the CCGCRV and Theil–Sen approaches. The mean annual growth rate of $CO_2$ over the 2011–2018 period is 2.5 ppm yr$^{-1}$ using the Theil–Sen method and 2.3 ppm yr$^{-1}$ using CCGCRV. This is consistent with the Mauna Loa global station rate, which is also 2.4 ppm yr$^{-1}$ on average for the period 2011–2018. It is stronger than the growth rate reported for the ZSF mountain station, 1.8 ppm yr$^{-1}$ over 1981–2016 (Yuan et al., 2019), and 2.0 ppm yr$^{-1}$ for the CBW station over 2005–2009 (Vermeulen et al., 2011). Such comparisons are only qualitative and must be used with caution, as the time periods considered are different. However, they suggest that the atmospheric $CO_2$ growth may speed up in the European mid-latitudes.

The OPE mean $CH_4$ annual growth rate over the 2011–2018 period is 8.8 ppb yr$^{-1}$ using CCGCRV and 8.9 ppb yr$^{-1}$ using the Theil–Sen method. It is slightly larger than the annual increase in globally averaged atmospheric methane from NOAA, which is 7.5 ppb yr$^{-1}$ over the 2011–2017 period. A slightly decreasing non-significant trend is seen for CO at OPE over the 2011–2018 period. This finding is consistent

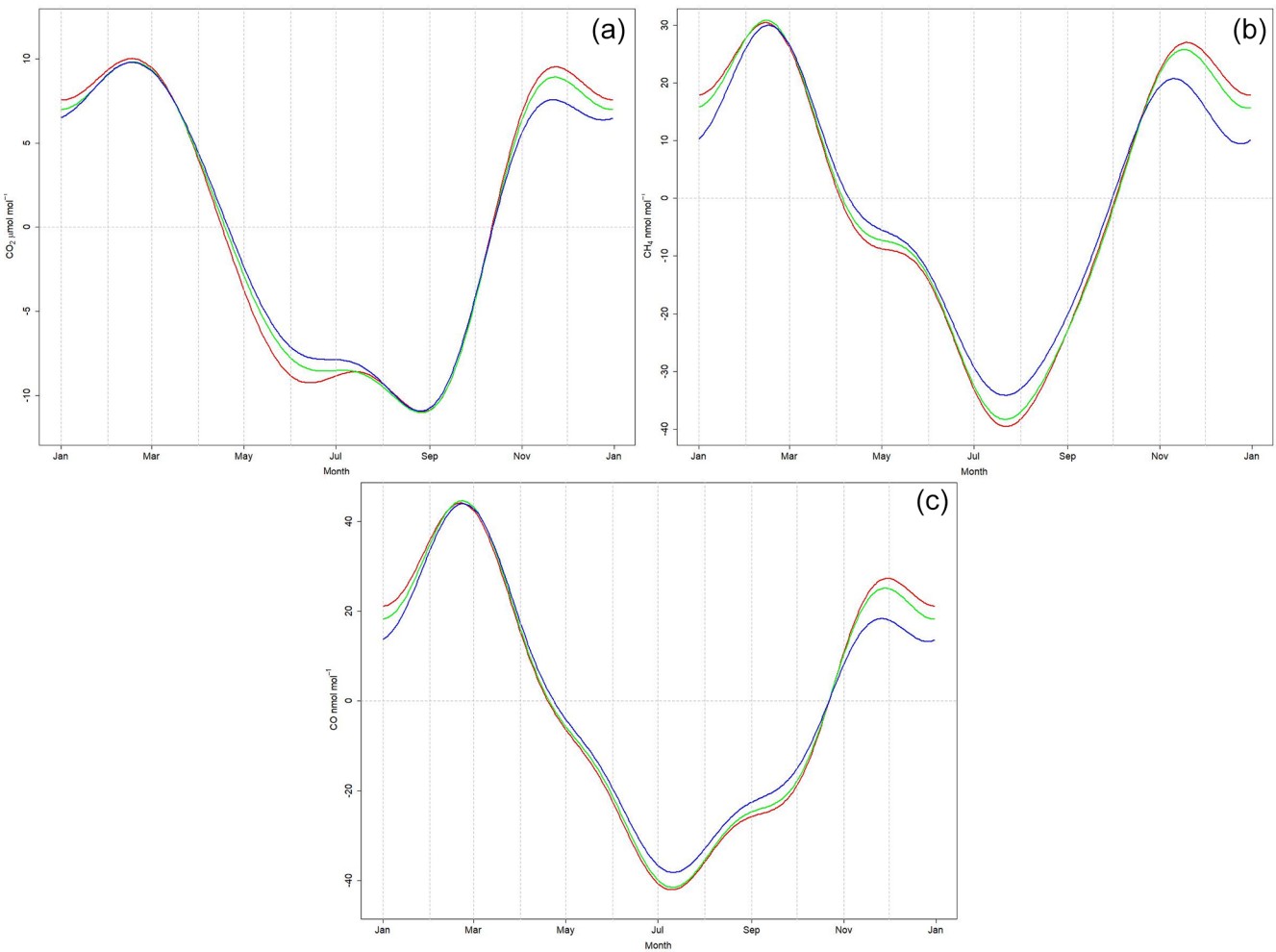

**Figure 11.** Mean seasonal cycles of the afternoon data at the three measurement levels (10 m in red, 50 m in green and 120 m in blue) for $CO_2$ **(a)**, $CH_4$ **(b)** and CO **(c)** computed over the 2011–2018 period using CCGCRV.

**Table 7.** Growth rates of $CO_2$, $CH_4$ and CO mole fractions at OPE 120 m level for the period 2011–2018 computed on the afternoon mean data using the CCGCRV and Theil–Sen methods. The 95 % confidence intervals are displayed for each compound and method.

| OPE-120m | $CO_2$ (ppm yr$^{-1}$) | $CH_4$ (ppb yr$^{-1}$) | CO (ppb yr$^{-1}$) |
|---|---|---|---|
| CCGCRV 2011–2018 | 2.35 (1.93; 2.77) | 8.85 (7.35; 10.34) | −0.22 (−3.9; 3.5) |
| Theil–Sen 2011–2018 | 2.54 (1.92; 3.28) | 8.91 (7.64; 9.96) | −0.49 (−1.71; 0.73) |

with recent observations in Europe and in the USA (Lowry et al., 2016; Novelli et al., 2003; Zellweger et al., 2009).

## 4.6 $CO_2$, $CH_4$ and CO residuals

We analysed the 120 m level residuals from the trend and seasonal cycle fitted curves with regard to air mass back-trajectories using the six clusters defined for the afternoon (see Fig. 2). Figure 12 shows the box plots of the residuals for each month and back-trajectory cluster. The box plot

displays the first and third quartiles and the median of the residuals along with the overall data extension.

The residuals of the three compounds are significantly stronger in the cold months than in the warm months. Clusters 5 (shown in blue) and 6 (in cyan) are associated with typical oceanic air masses with 96 h back-trajectories reaching far over the Atlantic Ocean. These air masses are associated with the lowest variability of residuals (smallest box plot extension). Negative residuals are noticed year-round for $CH_4$ and CO and during the cold months for $CO_2$ (positive during warm months). Clusters 1 (brown) and 2 (red) are associated

**Table 8.** Correlation coefficients between the compound residuals for each cluster, split between a warm period from April to September and a cold period from October to March.

| Cluster | 1 | | 2 | | 3 | | 4 | | 5 | | 6 | |
|---|---|---|---|---|---|---|---|---|---|---|---|---|
| Period | Warm | Cold | Warm | Cold | Warm | Cold | Warm | Cold | Warm | Cold | Warm | Cold |
| $CO_2/CH_4$ | 0.21 | 0.92 | 0.33 | 0.89 | 0.01 | 0.84 | 0.47 | 0.86 | 0.18 | 0.8 | 0.24 | 0.87 |
| $CO_2/CO$ | 0.16 | 0.91 | 0.4 | 0.87 | 0.24 | 0.85 | 0.52 | 0.91 | 0.24 | 0.74 | 0.24 | 0.78 |
| $CH_4/CO$ | 0.74 | 0.93 | 0.87 | 0.84 | 0.71 | 0.87 | 0.76 | 0.92 | 0.75 | 0.85 | 0.78 | 0.88 |

with southern and eastern trajectories. The associated residuals are much stronger and show large variabilities among the different synoptic situations with potential large deviations from the background.

Positive residuals are associated with cluster 2 year-round for $CH_4$ and CO and during the cold months for $CO_2$. Cluster 3 (orange) is associated with either negative or positive residuals for the three compounds. Cluster 4 (green) is characterised by relatively "stagnant" air masses with back-trajectories that do not extend far from the station in any particular direction. This type of air mass is associated with high residual variability for the three compounds during the cold period. The residuals can be either positive or negative and show large spreads among the situations.

Table 8 shows the correlation coefficients between the compound residuals for each back-trajectory cluster, split between a warm period from April to September and a cold period from October to March. During the warm period, the correlation coefficients between $CO_2$ and either $CH_4$ or CO residuals are low except for cluster 4. However, the correlation coefficients between $CH_4$ and CO are around 0.75 for each cluster. During the cold period, the correlation coefficients between residuals of the different compounds are high and significant for every type of back-trajectory. Similar seasonal patterns for the $CO_2$ and CO residuals and CO and $CH_4$ residuals were shown by Satar et al. (2016) in their 2-year analysis of the Beromünster tower data in Switzerland.

Such patterns suggest that, during the cold months, the variations in the three compounds are associated with the same anthropogenic processes convoluted through atmospheric dispersion. However, during the warm months, intraseasonal variations in $CO_2$ residuals may have different drivers than CO or $CH_4$ residuals, or their scale footprints are different. For example, natural biospheric contributions from different scales (local to continental) are larger for $CO_2$ during the warm months. Photochemical reactions are also much more activated during summertime. This result suggests that biospheric $CO_2$ fluxes may be the dominant driver of $CO_2$ intraseasonal variations during the warm period while anthropogenic emissions lead to intraseasonal variations in the three compounds during the cold period.

## 5 Conclusion

The OPE station is a new atmospheric station that was set up in 2011 as part of the ICOS Demonstration Experiment. It is a continental station sampling regionally representative air masses. In addition to greenhouse gases and meteorological parameters mandatory for ICOS, the station measures aerosol properties and radioactivity and is part of the regional air quality network. The GHG measurements are performed in compliance with the ICOS atmospheric station specifications, and the station was labelled by ICOS in 2017. We have presented the GHG measurement system as well as the quality control performed. Next, analysis of the diurnal cycles, seasonal cycles and trends were given for the GHG data over the 2011–2018 period. Finally, we analysed the compound residuals with regard to the air mass history.

The results of the monthly mean field CMRs and LTRs show that $CO_2$, $CH_4$ and CO measurements were compliant with the ICOS precision and reproducibility limit specifications except for CO during some period when spare instrument 187 was in operation. $CO_2$ and $CH_4$ measurement quality improved with time but not for CO. Biases were estimated on a regular basis with the station working standards and during Cucumbers intercomparison programmes. The station was also audited twice, just after its launch in 2011 and then in 2014. The audit results along with the routine quality control metrics such as CMR, LTR, and biases and the Cucumbers intercomparisons showed that the OPE station met the compatibility goals defined by the WMO for $CO_2$, $CH_4$ and CO most of the time between 2011 and 2018 (WMO, 2018). The station set-up and its standard operating procedures are also fully compliant with the ICOS specifications (Laurent et al., 2017).

The diurnal cycles of the three compounds show amplification of the vertical gradient at night mainly caused by the night-time boundary layer stratification associated with ground cooling and radiative loss. Minimum values are reached during the afternoon when vertical mixing is more efficient. In addition to this influence of the main atmospheric dynamics, diurnal cycles of surface emissions and of photochemical processes also play some role in the diurnal profiles of the three compounds. We focused on the afternoon data as we are interested in larger-scale processes. We computed the mean seasonal cycles of $CO_2$, $CH_4$ and CO. Relatively

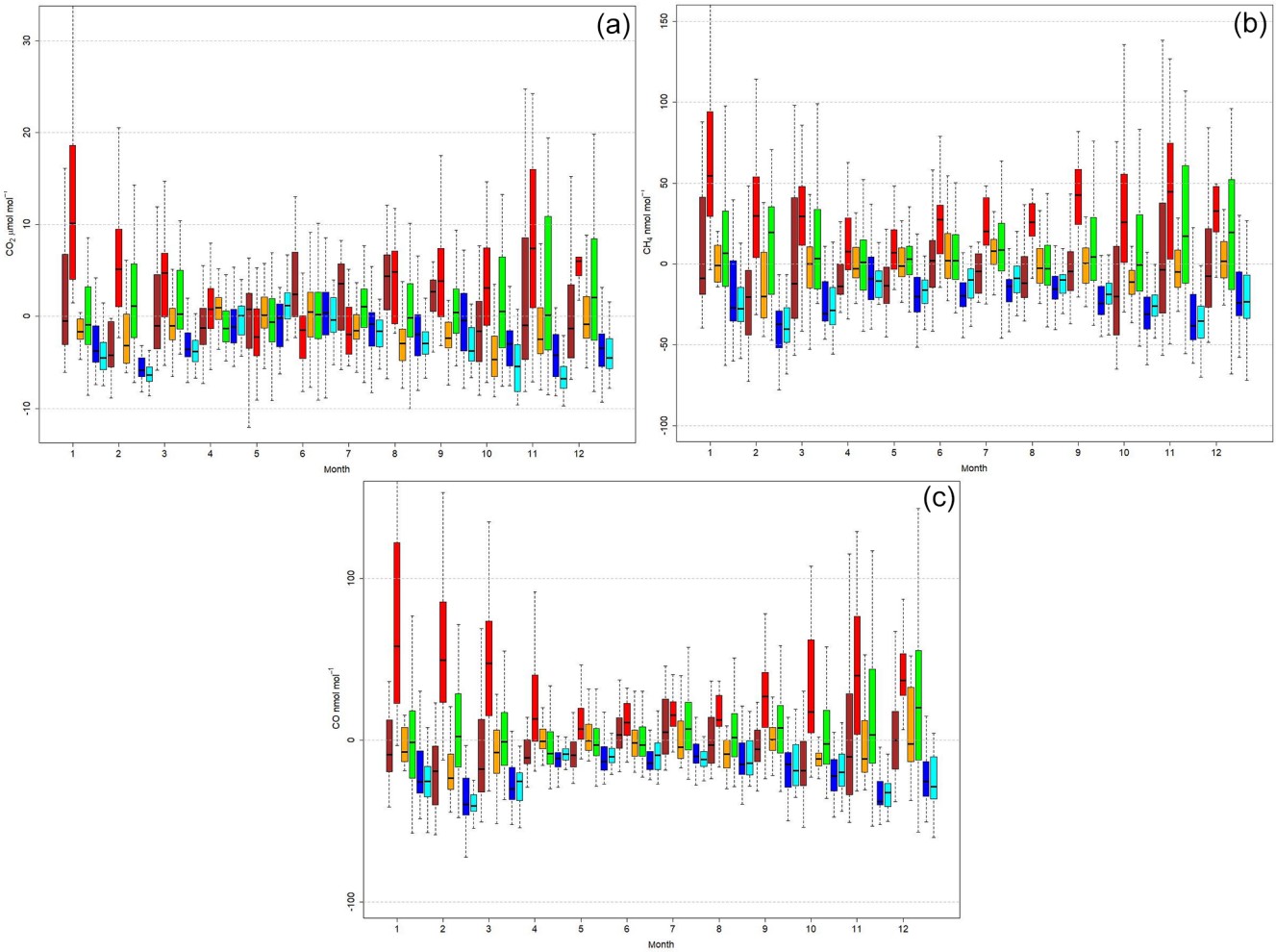

**Figure 12.** Seasonal box plot of the $CO_2$ **(a)**, $CH_4$ **(b)** and CO residuals **(c)** at OPE 120 m levels by cluster occurrence (cluster 1: brown; cluster 2: red; cluster 3: orange; cluster 4: green; cluster 5: blue; cluster 6: cyan) for the period 2011–2018.

strong positive trends were observed for $CO_2$ and $CH_4$ with a mean annual growth rate of 2.4 ppm $yr^{-1}$ and 8.8 ppb $yr^{-1}$ respectively for the period 2011–2018. No significant trend was observed for CO.

The residuals from the trends and seasonal cycles are much stronger during the cold period (October to March) than during the warm period (April to September). Our analysis of the residuals highlights the major influence of air masses on the atmospheric composition residuals. Air masses originating from the western quadrant with an Atlantic Ocean signature are associated with the lowest residual variability. Eastern continental air masses or stagnant situations are associated with larger residuals and high variability. The correlations between the compounds' residuals are also stronger during the cold period. Furthermore, there is no significant correlation between $CO_2$ and CO or $CH_4$ during the warm period.

*Data availability.* Data are available upon request to the corresponding author. Data are also available through the ICOS Carbon portal (https://doi.org/10.18160/CE2R-CC91, ICOS RI, 2019).

*Supplement.* The supplement related to this article is available online at: https://doi.org/10.5194/amt-12-1-2019-supplement.

*Author contributions.* SC, JH, LL, OL and MD performed the instrumental set-up and maintenance. SC, JH and OL carried out the data curation. MR supervised the station operations and provided suggestions for the data analysis, interpretation and discussion. JH prepared most of the figures. SC wrote the paper with contributions from all co-authors.

*Competing interests.* The authors declare that they have no conflict of interest.

*Special issue statement.* This article is part of the special issue "The 10th International Carbon Dioxide Conference (ICDC10) and the 19th WMO/IAEA Meeting on Carbon Dioxide, other Greenhouse Gases and Related Measurement Techniques (GGMT-2017) (AMT/ACP/BG/CP/ESD inter-journal SI)". It is a result of the 19th WMO/IAEA Meeting on Carbon Dioxide, Other Greenhouse Gases, and Related Measurement Techniques (GGMT-2017), EMPA Dübendorf, Switzerland, 27–31 August 2017.

*Acknowledgements.* The authors gratefully acknowledge the NOAA Air Resources Laboratory (ARL) for the provision of the HYSPLIT transport and dispersion model and READY website (http://www.ready.noaa.gov, last access: 8 November 2019) used in this publication. Samuel Hammer from Heidelberg University and Hermanni Aaltonen from FMI are thanked for their efforts during the OPE station audits. Staff from IRFU-CEA are acknowledged for their contribution to the station's initial design and installation. We also thank staff from SNO-ICOS-France and from the ICOS Atmospheric Thematic Center for their technical support. The authors would like to thank the editor and two anonymous referees, who provided valuable suggestions and constructive comments to improve the paper.

*Review statement.* This paper was edited by Christoph Zellweger and reviewed by two anonymous referees.

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
