# Peer review of "Continuous atmospheric CO2, CH4 and CO measurements at the Observatoire Pérenne de l'Environnement (OPE) station in France from 2011 to 2018"

_Atmospheric Measurement Techniques, 2019_

## Referee Comment (RC1) · Anonymous Referee #1 · 5 Jun 2019

**Overview:**

Conil, et al., have submitted a manuscript for publication detailing continuous greenhouse gas (CO2, CH4, CO) observations at the Observatoire Pérenne de l'Environnement (OPE) station, France. The focus of the manuscript is on multi-analyser (and multi-height sampling system) performance over a 7-year period and subsequent analysis of the resultant quality controlled timeseries. Diurnal cycles, seasonal cycles and inter annular trends are calculated and commented upon in context of air mass back trajectory analysis. The OPE station is an important component of the ICOS network providing high quality data. Such data uses will include national and pan-national 'top down' GHG inventory emission monitoring.

The novelty of this manuscript is that this is the first time that the OPE station instrument performance has been explicitly evaluated along with a preliminary analysis of data. The OPE continuous greenhouse gas observations are conducted under the auspices of the ICOS in situ measurement framework, hence all measurements, performance metrics, auditing techniques and data selection/filtering at OPE must meet ICOS standards. OPE data is centrally processed at the ICOS-ATC. As such, the authors defer to published work by Hazan et al., AMT, 2016 to define OPE station data calibration and quality assurance procedures, thus the manuscript is the standard combination of site and meteorological descriptions, instrument performance and time series evaluation, but with a very minimal section on measurement calibration and data selection filtering.

The manuscript content is in the scope of the AMT journal. This research will be a welcome addition to already published ICOS network literature and long term in situ analyser performance. Unfortunately, the manuscript is let down in multiple critical areas and I do not recommend publication until the issues listed below are addressed; either fixed or with a sufficient logical rebuttal. The language and structure of the manuscript can be improved. Scientific methods and assumptions need to be clarified. I have concerns (or maybe just a lack of detail) about the methodology of combining multiple instrument data into a single timeseries. There is incomplete analysis of datasets (lack of uncertainty estimates). There needs to be more collaborative evidence from peer reviewed literature to support conclusions deduced from analysis.

**Specific comments:**

S1/ AMT English guidelines and house standards: A major draw-back of the submitted manuscript is that I do not believe the grammar meets the standard required for publication in AMT. The authors https://www.atmospheric-measurementare referred to AMT guidelines: techniques.net/for authors/manuscript preparation.html. There are many instances of incorrect grammar use, such is non-defined subjects (nouns), use of colloquialisms, non-defined acronyms along with simple grammatical errors. All such instances need to be corrected. This is no reflection on the quality of the science presented and doesn't detract (only distracts and introduces ambiguity) from the novelty and importance of the presented subject matter along with the effort the authors have already put into the manuscript. As an example, the majority of the first 18 technical comments (see below) are related to grammatical errors in the abstract and first section of the manuscript. For the remainder of the manuscript review correction of such grammatical errors will be left out (to speed up the review), and only commented upon if scientific clarity is required.

S2/ The term "Afternoon mean residuals" is introduced in the abstract and section 2.2, but the term is not defined until section 4.5. A more detailed description is required early in the manuscript, or a reference to later sections (i.e. see section XX for the definition of 'Afternoon mean residuals").

S3/ Page 4, Line9 (pg4, L9). The criteria used to define the six clusters needs to be included.

S4/ The section detailing the calibration strategy: pg 7, L5 to pg 8, L2 needs to be reorganised. The section starts by explaining the cylinder measurements, then details the reference scale then back to the routine operating sequence (including flushing). I suggest the routine operating sequence (sample measurements, flushing, injections etc) be moved to the start, followed by the calibration (this will logically allow how the calibration cycles fit into the overall measurement scheme ) then describe the reference scale. Maybe include a table like Table 3 from Hazan, et al., AMT 2016 (H16) but specifically for the OPE station operation.

S5/ Concerning the performance and standard cylinders (pg 7 L10 to L15). As the manuscript reads, the measurements made pre and post March 2016 are on difference scales for some species. Are measurements all recalibrated onto the same scale (per species) later? The details are not clear if this is done or not.

S6/ The paragraph starting pg 8 L3 concerning the 14CO2 measurements seems outside the scope of this manuscript. Should it be removed along with the non-continuous GHG measurements listed in Table 1? It seems the manuscript content is solely concerned with the description and data interpretation of the continuous GHG analysers. The scope of the manuscript is stated on pg 2, L26: "Describe the OPE station and measurement system. Present its performance..." I think the scope needs to change to only include the continuous GHG systems, or the manuscript expanded to include performance of all instruments...which could be a lot of work.

S7/ Table 1 has columns of identical naming, i.e. period 1. I assume these are the start and stop dates for each period? Column naming needs to be tidied up. Even if this was done, it is hard to understand. Would the authors consider replacing the period columns with a time line graph, with each instrument a separate bar? This way it would be easy to see dates and overlapping periods.

S8/ Section 2.4 should be renamed 'data processing', (currently section 2.5). The first paragraph in Section 2.5 needs to be put in this, along with the current section 2.4 as data processing should be explained before combining any datasets. The second and third paragraphs in the current section 2.5 need to be moved to section 4.2 as it deals with analysis of a subsection of data. Current section 2.5 is now not needed.

S9/ For section 2.4, there is lot of broad qualitative reasoning for instrument issues. I recommend the authors make more extensive use of H16 by referencing the types of QA/QC practise used and provide a quantitative statistical summary of the OPE site, like that for OPE in table 6 of H16. On pg 10, L18 it states, "Raw data is flagged using a set of parameters defined for the station and instrument". This is where a quantitative statistical summary for OPE specifically would be useful.

S10/ Table 2 is very complicated and hard to understand. The caption is not helpful. Possibly make a bar plot, as in comment S7, or a table per species.

S11/ Combination of instrument time series. Please detail how priority is set, the instrument with the best precision or 'best' QC/QA? Is there an ICOS procedure to follow for the combination of different instrument timeseries at one location? I find figure 4 a very important piece of information in this manuscript. The current figure does not relay much information. It effectively is central to the OPE total timeseries, as such it would be very helpful to either change figure 4 to display statistics, such as box whisker plots or include another table with the bias and spread of instrument overlap differences. Something akin table 4 in Schibig, et al. (2015). In the final instrument combined time series are the time periods of instrument overlap where there are large differences which instrument is kept? Or should such a disagreement exclude both measurements?

S12/ There is no mention of the GAW-recommended compatibility limits (GAW, 2011) in section 3 (it is mentioned in the travelling audit section, pg 16, L30). The authors may want to state the GAW compatibility limits and how OPE CMR and LTR compare to these (such in the paragraph starting at pg 12, L15). Does ICOS have a precision and reproducibly limits that needs to be reached? If so this could also be stated and OPE CMR and LTR statistics compare to this guideline instead of the GAW limits.

**GAW: Report no. 194, 15th WMO/IAEA Meeting of Experts on Carbon Dioxide, Other Greenhouse Gases and Related Tracers Measurement Techniques, Geneva, WMO/TD-No. 1553, 2011.**

S13/ CMR monthly means of the time series. Again, a very important part of the manuscript. CMR is related to single instrument performance. Calculating and displaying the CMR of combined instruments does not make sense and contradicts the definition of how the time series is constructed, in the sense that data selection is based upon instrument priority, and exclusion of the lower priority instrument data (pg 9, L2)? A combined CMR in Figure 5 (example 379:187) implies that the timeseries includes all overlapping measurement data. Is this correct? Does ICOS allow this practise? If so, then CMR calculation of a combined dataset should not be performed. I suggest that CMR should be calculated for each individual instrument to be displayed in figure 5. This also applies to LTR statistics in figure 6.

S14/ Pg 12, L3, "The time series of CO's CMR o are not shown as the intrinsic properties of the Picarro and Los Gatos Research analysers are very different making it difficult to compare on a same plot.". The performance of the instruments is central to this manuscript; thus, I think it is very important to also present the CO CMRs. CO LTRs are displayed in figure 6. The CO CMRs for the Picarro and Los Gatos can be displayed on separate plots.

S15/ Table 4. In both Picarro's (187 and 728), LTR is significantly less than CMR. There is no mention of this, or interpretation, as in principle LTR (reproducibility) should be greater than CMR (repeatability). Could the author please comment on this. Pg 15, L11 discusses the Los Gatos instruments but neglects to mention which species they are talking about.

S16/ Pg 13, L14: "These two types of analysers have very different internal properties making it difficult to show direct comparison.". I disagree with this comment. CMR and LTR can be directly compared and are defined to be independent of instrument internal properties. This is the idea behind using such statistics. Table 4 indicates that instruments 80 and 478 have better CMR and LTR than instruments 187 and 728.

S17/ Table 2 shows that the combined times series of CO includes measurements from all four instrument timeseries. This means that the CMR and LTR of the timeseries will have step functions. This should be mentioned in the manuscript (indirectly alluded to at pg 15, L8), preferably referencing H16 (as to how uncertainty estimates are delivered in the end user database).

S18/ Pg 15, L25 to pg 16, L11. Just a comment: The audit shows differences. Was there a change in OPE operation due to the audit results?

S19/ Figure 7, A box whisker plot would convey the target tank statistics a lot clearer with a box whisker plot per tank, per instrument. The cucumber tanks can be left as individual points.

S20/ Pg 18, L7. "A trend may be present". Yes, this is interesting, firstly I thought there was a clear trend, but on reflection there could be a step change at each tank. If the time series is a combination of multiple instrument datasets, then could this be the cause of a possible step change? Would the authors like to comment on possibilities of a continual trend or a series of step changes?

S21/ Section 4: Results. The first paragraph in this section mentions that general characteristics will be investigated, then diurnal cycles. There also is a need to state that seasonal cycles and long-term trend analysis will also be analysed and commented upon.

S22/ Section 4.1: General characteristics. Most of this section is about vertical concentration gradients thus should this section be called vertical concentration gradients (or something similar). If this title change is made then 'general characteristics' details can be moved to the appropriate section: diurnal, seasonal or long-term trend. There is also no commentary of the OPE vertical gradients in relation to other tall tower measurements in the same region (or Europe as a whole). Is the drawdown seen at OPE like other measurements? Is it anomalous? This section could use a few more references to contemporary literature to put OPE measurements in context.

S23/ Figure 9: there are no uncertainty, or spread, bars on these plots. Such uncertainty or spread is critical in such plots and must be displayed.

S24/ Figure 9: The caption states that the data is normalised to the 120 metre inlet height measurements. Why is this done? I cannot see the reason why. Wouldn't it be better to display the actual non-normalised data? Maybe I am misinterpreting.

S25/ Figure 9. Are the mean diurnal cycles deseasonalised and detrended? If so (or not) then it should be stated.

S26/ Section 4.1. There is no mention of any diurnal cycle in wind direction or speed. Are night time inversions seen? Is the diurnal cycle in  $CO_2$ ,  $CH_4$  and CO affected by such inversions or windy nights?

S27/ Section 4.2 As stated in prior comments (S8), pg 11 L3 to L12 should be moved to section 4.2.

S28/ Section 4.2 should be renamed to something other than the generic title of "data selection and time series analysis", as the section is predominantly concerned with well mixed boundary layer conditions. Data selection is a too generic term. The section should state that data is filtered to represent a well-mixed boundary layer, also state that this filtered data is to be used in seasonal and trend analysis.

S29/ The 'openair package' and the 'theilsen method' need referencing.

S30/ In the CCGCRV algorithm please specify how was the npoly and nharm variables are set, I.e. using a geophysical basis or iterative attempts to get the best fit?

S31/ Pg 21, L22. Comparison of CCGCRV residuals with REBS. The sentence on this line states a comparison was made, but no mention of any results of this 'qualitative' comparison. If the comparison was important then results should be mentioned, else maybe leave out the REBs comparison.

S32/ Figure 10. Like Fig 9 comments, no 'spread' (1-sigma?) bars for each month. These need to be included. The caption should also state if the seasonal cycles are detrended or not.

S33/ As in section 4.1, section 4.3 does not mention the seasonal cycle in context of any prior studies. Is the OPE station seasonal cycles anomalous or what is expected. The authors need to put their results into such context.

S34/ Pg 23, L18: "We analysed the residuals from the trend...". Residuals from which measurement height? Could the specific height be stated, or all three? (I'm sure it's 120m but should be explicitly stated).

S35/ Table 6. Uncertainty estimates are needed for all calculated trends parameters. Unlike previous sections, the OPE trends are compared to other sites. W. But no mention of the comparisons in respect to OPE or other station trend uncertainties. Please rectify.

S36/ Figure 11. What is OPE level 3? I gather the 120m height? Maybe remove references to level 3?

S37/ Pg 25, L23. "We presented the GHG measurement system as well as the quality control performed". Quality control (QC) for OPE was not presented. The QC method used was referenced to H16 and a qualitative description of filtering parameters and issues where given. Explicit OPE filtering diagnostics were not displayed. As stated in S9, the authors already have such statistics available through the ATC processing and should be easily incorporated into the paper.

S38/ Section 5 Conclusion: GAW and/or ICOS compatibility limits should be mentioned and referenced when discussing OPE CMR and LTR, travelling standard and target tank results.

**Technical comments (no particular order):**

T1/ Title: OPE full name should be used.

T2/ In the abstract provide the station coordinates as at pg 2, L30.

T3/ Pg 1, L10, 'several' should be replaced with the exact number.

T4/ Pg 1, L13: 'Thanks' replaced with 'Using the'

T5/ Pg 1, L16, Growth rates need uncertainty estimates.

T6/ Pg 1, L18: 'Afternoon mean residuals", residuals of what?

T7/ Pg 1, L19, what are warm and cold periods? Not defined yet.

T8/ Pg 1 L24, 'largest climate change contributor' can be taken out.

T9 / Pg 1, L26, 'High' placed with 'increased'.

T10/ Pg 1, L27, 'Those' what is this referring too?

T11/ Pg 2, L1, throughout the manuscript are the characters: '<<' and '>>'. These need to be removed.

T12/ Pg 2, L18, 'responsible of the' should be 'responsible for the'

T13/ Pg 2, L19, 'Andra' and 'LSCE' have not been defined. First time these are mentioned, so should be the full title.

T14/ Pg 2, L19. 'OPE' needs to be defined. The full name is only found in the abstract.

T15/ Pg 2, L20. 'It is a....', there is no subject. This should be replaced with 'OPE'

T16 Pg 2, L21, 'actual' not needed.

T17/ Pg 2, L27, 'draw' is a colloquialism and should be avoided.

T18/ Pg 3, L10. What is a "complete set"? A reference to a definition of "complete set" is required. I assume its defined by ICOS.

T19/ Pg 3, L12, ACTRIS and AERONET acronyms need full names.

T20/ Figure 1. Symbol (green house) for OPE station in panel 1 needs to be stated in the legend.

T21/ Pg 3, L23, A reference for 'ICOS AS specifications' is required. Should 'AS' be replaced with 'Atmospheric stations'?

T22/ Pg 4, L6, "96h back trajectories were computed for the OPE station top level (120m) using the NCEP reanalysis fields and HYSPLIT model every 6 hours". I think what the authors mean is "96h back trajectories were computed for the OPE station top level (120m) using the NCEP 6-hourly reanalysis fields and the HYSPLIT model" ...not reanalysis every 6 hours...

T23/ Pg 5, L3: "It was built in order to comply with the Atmospheric Station class 1 stations specifications from ICOS". This needs a reference, or is the reference to T21 sufficient?

T24/ Pg 5, L4: "It includes several continuous analysers for the main GHG CO2, CH4 and N2O, a manual flask sampler as well as specific analysers or samplers for tracers such as radon, CO and 14CO2.". I find it unclear it this refers to the OPE station specifically, or ICOS AS stations in general. Do all ICOS AS stations need serval analysers or just the OPE site? For completeness:  $14CO_2$  should be  $^{14}CO_2$ .

T25/ Figure 3. This is a detailed schematic. A legend with symbol definitions is required, or components labelled. The figure caption should state this is only for the continuous GHG analysers (it does not include flask sampling details).

T26/ Pg 5, L11: "using 0.5 inches outer diameter Dekabon tubings". The inner diameter is the important specification. Could the inner diameter please be stated?

T27/ Pg 5, L13: A reference for the 14CO2 system should be included (if this section kept)

T28/ Pg 5, L18: "At each level, the continuous GHG monitoring system air is flushed from the tower using three flushing pumps". This is unclear. Is it that each line at each level is permanently attached to an independent flushing pump? Could this please be made clearer in the manuscript.

T29/ Pg 5, L23: Could the 'pre-drying' fridge temperature be stated.

T30/ Pg 5, L24: Are the cryo-traps and ethanol bath the same thing? Reading the manuscript, it portrays a cryotrap is in series after the ethanol bath.

T31/ Pg 5, L25 and Figure 3. The Picarro G2301 is given in figure 3, but not mentioned in section 2.3. Please add a sentence or paragraph explaining the G2301 and operation of it.

T32/ Pg 5, L27. PLC: could the make and model be listed.

T33/ Pg 6, L1. Could the model of the Vici Valco be given for completeness?

T34/ Pg 6, L3. 'Avoid' should be replaced with 'reduced', or 'significantly reduced'

T35/ Pg 6, L3. The statement 'According to ICOS rules' needs to be referenced.

T36/ Pg 6, L4. 'global' should be replaced with 'total system'.

T37/ Pg 7, L13, What is 'CAL'? The full title should be given in first instance.

T38/ Pg 11, L18. "Up to now". 'Now' is subjective, please use dates.

T39/ Pg 12, L19. STR is not defined, please define.

T40/ Pg 13, line 4. "MLab" is not defined prior to use, please define.

T41/ Table 3. The authors may want to combine tables 3 and 5, like in table 4.

T42/ Pg 14, L7. "The Figure 6 shows the monthly mean field LTR of the merged time series using the different instruments and sampling systems. This figure shows the uncertainties of the data related to the analysers (not the sampling systems)". The second sentence contradicts the first sentence. Since the tank gas is injected into the analysers downward of the atmospheric sampling system then I suspect the second sentence is more correct. Please correct the statement above.

T43/ Pg 15, L13: "Corrections for these temperature induced biases implied the use of a working standard quite frequently". Please define 'quite frequently'.

T44/ Pg 16, L6: Define "poor performances". A list of "specific hardware problems" would be help help i.e. "specific hardware problems (e.g., XXX, XXXX)"

T45/ Pg 16, L12. "Piccaro G2401 travelling instrument". Please state who owns this instrument. I assume the Finnish Met ICOS mobile lab?

T46/ Pg 16, L13. "FTIR performance was not yet optimised". Remove 'yet'. What is meant by 'not yet optimised'.

T47/ Pg 16, L19: "lightning" to "lightning strikes"

T48/ Pg 16, L21: "Degrading instrument performance", do you mean a decrease in precision or accuracy, or both?

T49/ Pg 16, L21: "The ambient air comparison", of what gases. I presume all (CO2, CH4, CO)? if so, it should be stated.

T50/ Pg 18, L12. "over mid latitudes", just mid latitudes?

T51/ Pg 20, L3. "vertical gradients of CO are much stronger in winter and weaker in summer", can this please be quantified.

T52/ Pg 20, L4. "The CO lifetime...". This sentence needs a reference.

T53/ Pg 21, L23, what are REBs? The acronym should be defined before use.

T54/ Pg 23, L12. The sentence "This finding is consistent with recent observations in Europe and in the US" needs to be referenced.

T55/ Figure 11 caption needs to state the box and whisker plot definitions. The sentence on pg 23, L19, "The figure 11..." can be moved to the figure 11 caption. 'Overall data extension' should be replaced with 'overall data range'.

T56/ Pg 25, L6: "...between  $CH_4$  and CO are around 0.75", adding a range would be good, i.e. "correlations between XX to XX are found".

T57/ Pg 25, L7. "large" and significant" need to be defined. Maybe better to state the correlations are greater than XX.

T58/ The last three sentences starting pg 25, L13 all need to be referenced.

T59/ Pg 25, L20. "sampling air masses influence with regional footprints" is not needed. All stations have this, nothing specific to OPE.

T60/ Pg 25, L23. What is 'ICOS-ERIC', please define and what does 'labelled' mean?

T61/ Pg 25, L18. "no significant correlation", define 'no significant'.

---

## Referee Comment (RC2) · Anonymous Referee #2 · 2 Jul 2019

General comments:

The authors presented 8 years of station data, from the Observatoire Perenne de l'Environnement (OPE), which is situated on the eastern edge of the Paris Basin in NE France. As such, this regional station represents continental rural background measurements to the ICOS network and contributes valuable data to link the existing oceanic and urban observation sites. With this study the authors also successfully showed how to interpolate and analyse composite merged data sets, obtained from various sampling analysers in order to comply with stringent ICOS data quality objectives. The paper as a whole is well written and presented and met the objectives set

out in the introduction.

Specific comments:

Page 10, line10: Prior to this, the authors described differences (in afternoon) between instruments at the same intake height... this was then followed by a remark that "Schibig et al..." found some similar large deviations at their site. Perhaps a better explanation is needed here? Or a table listing the authors' observations in context with other literature reported differences? As it currently reads – it just seemed a bit out of context to me.

Page 12, Lines 15-20: Please put this info in a table format – it makes the intercomparison of the different parameters much easier to read and compare.

Page17, Figure7: improve y-axis font (make larger); CO bias graph – improve scale to say $\sim$2 nmol.mol-1 intervals to show WMO compatibility;

Page 23, Lines6-8: I understand the point being made by the authors (i.e. a comparison of observed growth rate at OPC against other nearby sites...) but perhaps a better explanation is required when this is compared to Zugspitze? (the Zugspitze growth rate comparison is based on a 1981- 2016 determination...) and Cabauw on a 2005 – 2009 value for that matter. My question being – Can one draw any useful comparison across such large timescale differences?

Technical corrections/ comments:

Most of these corrections are as a result of the authors not being English first language speakers and are minor language issues...

Page 1, line28: rephrase sentence..."Remote and mountain atmospheric measurements..."

Page 5, line7: rather use singular for (1) "measurement" and not "measurements"; (2) "ambient air sample" and not "samples"

Page 5, line9: replace "station's " with "stations"... replace "on" with "in"

Page 6, line8: replace "went first" with either "first went" or "was subjected to..."

Page 6, line10: replace "informations" with "information"

Page8, line16: replace "lightnings" with "lightning"

Page 8, line 19: fan, ....) add "etc." {et cetera}

Page 9, line9: remove double space after "...efficiency)"

Page 10, line4: use plural "sources"

Page11, line29: use singular "measurement"

Page16, Line10: use singular "measurement"

Page18, Line1: replace "to" with "in"

Page 18, Line2: ditto - replace "to" with "in"

Page21, Line30: use plural "dynamics"

Page 21, Line31: add "it" to "...seasonal scale make difficult..."

Page26, Line10: Rephrase sentence "Interested on larger... data"

Page28, Line29: Please check and ensure that the references comply to the journal's requirements "Lowry, D. et al..." Full reference required?

---

## Author Comment (AC1) · 28 Aug 2019

**Reply to the reviewers**

We would like to thank the anonymous referees for their constructive comments which helped us to improve the manuscript. We adress their individual comments below. In the following, referee's comments are given in bold, author's responses in plain text. Suggested new text is quoted in italics.

*Anonymous Referee #1*

Overview:

**Conil, et al., have submitted a manuscript for publication detailing continuous greenhouse gas ($CO_2$, $CH_4$, CO) observations at the Observatoire Pérenne de l'Environnement (OPE) station, France. The focus of the manuscript is on multi-analyser (and multi-height sampling system) performance over a 7-year period and subsequent analysis of the resultant quality controlled timeseries. Diurnal cycles, seasonal cycles and inter annular trends are calculated and commented upon in context of air mass back trajectory analysis. The OPE station is an important component of the ICOS network providing high quality data. Such data uses will include national and pan-national 'top down' GHG inventory emission monitoring.**

**The novelty of this manuscript is that this is the first time that the OPE station instrument performance has been explicitly evaluated along with a preliminary analysis of data. The OPE continuous greenhouse gas observations are conducted under the auspices of the ICOS in situ measurement framework, hence all measurements, performance metrics, auditing techniques and data selection/filtering at OPE must meet ICOS standards. OPE data is centrally processed at the ICOS-ATC. As such, the authors defer to published work by Hazan et al., AMT, 2016 to define OPE station data calibration and quality assurance procedures, thus the manuscript is the standard combination of site and meteorological descriptions, instrument performance and time series evaluation, but with a very minimal section on measurement calibration and data selection filtering.**

**The manuscript content is in the scope of the AMT journal. This research will be a welcome addition to already published ICOS network literature and long term in situ analyser performance. Unfortunately, the manuscript is let down in multiple critical areas and I do not recommend publication until the issues listed below are addressed; either fixed or with a sufficient logical rebuttal. The language and structure of the manuscript can be improved. Scientific methods and assumptions need to be clarified. I have concerns (or maybe just a lack of detail) about the methodology of combining multiple instrument data into a single timeseries. There is incomplete analysis of datasets (lack of uncertainty estimates). There needs to be more collaborative evidence from peer reviewed literature to support conclusions deduced from analysis.**

The authors would like to thank the anonymous referees #1 for her/his constructive general comments. We worked on the structure as well as the language to improve the manuscript. We added some new texts to present the merging time series procedure. We also introduced some additional references as suggested. The details are presented below regarding each specific comments)

Regarding the lack of uncertainty estimates, we agree with the referee that it is an important matter. However, a full assessment of time varying uncertainties remains a real challenge for our community, which has not yet succeeded in proposing a robust and operational methodology. An ICOS working group is dedicated to make progress on this issue, and the outcome of the discussions will be presented in a future paper. For the present work we are convinced that the QA/QC metrics, as well as the intercomparison experiments results provide valuable qualitative informations about the data quality at OPE Consequently we consider that the full uncertainty estimate is beyond the scope of this paper.

**S1/ AMT English guidelines and house standards: A major draw-back of the submitted manuscript is that I do not believe the grammar meets the standard required for publication in AMT. The authors are referred to AMT guidelines: https://www.atmospheric-measurement-techniques.net/for_authors/manuscript_preparation.html There are many instances of incorrect grammar use, such is non-defined subjects (nouns), use of colloquialisms, non-defined acronyms along with simple grammatical errors. All such instances need to be corrected. This is no reflection on the quality of the science presented and doesn't detract (only distracts and introduces ambiguity) from the novelty and importance of the presented subject matter along with the effort the authors have already put into the manuscript. As an example, the majority of the first 18 technical comments (see below) are related to grammatical errors in the abstract and first section of the manuscript. For the remainder**

of the manuscript review correction of such grammatical errors will be left out (to speed up the review), and only commented upon if scientific clarity is required.

We altered the manuscript so that acronyms are defined the first time they are used. For the correction of the simple grammatical errors and colloquialisms, the paper was corrected by an english native speaker.

**S2/ The term "Afternoon mean residuals" is introduced in the abstract and section 2.2, but the term is not defined until section 4.5. A more detailed description is required early in the manuscript, or a reference to later sections (i.e. see section XX for the definition of 'Afternoon mean residuals").**

We have now rephrased sentence in the abstract and on line 12 page 4 as follows:

*The afternoon mean residuals (defined as the differences between midday observations and a smooth fitted curve)*

**S3/ Page 4, Line9 (pg4, L9). The criteria used to define the six clusters needs to be included.**

The criteria used to select 6 clusters is based on the Total Spatial Variance computed by the HYSPLIT clustering tool. It is a metric describing the sum of all the cluster spatial variances. For large number of clusters it is quite low and it increases slowly as the clusters number decreases. At some point, the Total Spatial Variance starts to increase significantly meaining that disparate clusters are combined together. This number of clusters is selected as the optimal cluster number sorting similar trajectories. We added the following sentence on page 5 line 2:

*Based on the total spatial variance (TSV) metric, describing the sum of the within cluster variance, the optimal number of clusters was six (lowest number with a small TSV). The TSV plot is shown on the figure S1 in the supplementary material.*

**S4/ The section detailing the calibration strategy: pg 7, L5 to pg 8, L2 needs to be reorganised. The section starts by explaining the cylinder measurements, then details the reference scale then back to the routine operating sequence (including flushing). I suggest the routine operating sequence (sample measurements, flushing, injections etc) be moved to the start, followed by the calibration (this will logically allow how the calibration cycles fit into the overall measurement scheme ) then describe the reference scale. Maybe include a table like Table 3 from Hazan, et al., AMT 2016 (H16) but specifically for the OPE station operation.**

We reorganised the section 2.3 following the reviewer #1 suggestion. A table S1 like Table 3 from Hazan, et al. (2016) was included in the supplementary materials to describe the OPE routine measurements sequence.

**S5/ Concerning the performance and standard cylinders (pg 7 L10 to L15). As the manuscript reads, the measurements made pre and post March 2016 are on difference scales for some species. Are measurements all recalibrated onto the same scale (per species) later? The details are not clear if this is done or not.**

All the measurements were recalibrated on the same scale per species by the ATC. We added the following sentence on p8 line 8:

*All the measurements data presented here were recalibrated on those later scales.*

**S6/ The paragraph starting pg 8 L3 concerning the 14CO2 measurements seems outside the scope of this manuscript. Should it be removed along with the non-continuous GHG measurements listed in Table 1? It seems the manuscript content is solely concerned with the description and data interpretation of the continuous GHG analysers. The scope of the manuscript is stated on pg 2, L26: "Describe the OPE station and measurement system. Present its performance..." I think the scope needs to change to only include the continuous GHG systems, or the manuscript expanded to include performance of all instruments...which could be a lot of work.**

We agree with this comment. We removed the corresponding sentence about $^{14}CO_2$ measurements and focus the scope in the introduction. We rephrased the last sentence of the introduction page 2 line 32 :

*The main objectives of this paper are to describe the OPE monitoring station, the continuous GHG measurements system, to present its performance and to draw some results from the first eight years of continuous operations*

**S7/ Table 1 has columns of identical naming, i.e. period 1. I assume these are the start and stop dates for each period? Column naming needs to be tidied up. Even if this was done, it is hard to understand. Would the authors consider replacing the period columns with a time line graph, with each instrument a separate bar? This way it would be easy to see dates and overlapping periods.**

We included a time line graph as suggested by the reviewer and the table was moved to the supplementary materials. The column names of the Table were modified to include the start and end of each period.

**S8/ Section 2.4 should be renamed 'data processing', (currently section 2.5). The first paragraph in Section 2.5 needs to be put in this, along with the current section 2.4 as data processing should be explained before combining any datasets. The second and third paragraphs in the current section 2.5 need to be moved to section 4.2 as it deals with analysis of a subsection of data. Current section 2.5 is now not needed.**

We modified the section 2.4 as suggested by the reviewer and removed the section 2.5, moving the first paragraph to section 2.4 and some other parts to section 4.2.

Section 4.2 is now written as below :

*Our aim in this paper is to draw the general behaviours of the major GHG at the station focusing on relatively large scale. The station hourly time series exhibit strong variability from hourly to interannual time scales. These variations may be related to meteorological and climate changes, and to sources and sinks variations. We are mostly interested in the regional signatures at scales that can be approached by the model inversion and assimilation framework. For this reason we want to isolate from the time series and data aggregation the situations where the local influence is dominant and is shadowing the regional signature. We then need to define the background signal on top of which the regional scale signal is added.*

*Such local situations and background definitions may be extracted purely from time series analysis procedures, or may be constrained on a physical basis. The main difficulty is to correctly define the baseline signal of the measured time-series and to adequately flag local spikes. El Yazidi et al. (2018) have assessed the efficiency and robustness of three statistical spikes detection methods for $CO_2$ and $CH_4$ and have concluded that the two automatic SD and REBS methods could be used after a proper parameters specification. We used the El Yazidi et al. (2018) method on the composite merged minute time series to filter out « spike » situations. From this despiked minute dataset we built hourly means, which were used to analyse the diurnal cycles. Focusing on data with regional footprints, we selected only afternoon data with low hourly variability when the boundary layer is larger and the vertical mixing is more efficient. We excluded data showing large variations by using the minute standard deviations. Hourly data with minute standard deviations larger than three interquartile range computed month by month were excluded from the afternoon mean, leading to a rejection of between 2.9 % and 4.2% of the hourly means of the $CO_2$, $CH_4$ and CO.*

*We then used the CCGCRV curve fitting program program from NOAA (Thoning et al., 1989) with the standard parameters set (npoly=3, nharm=4) to compute the mean seasonal cycles and trends for the three compounds. CCGCRV results were compared with similar analysis performed with the openair package of R for the seasonal cycle and the trend using the Theilsen method. These seasonal cycle and trend components of the time series are dominated by large-scale processes. In addition strong intra-seasonal variabilities are observed that are related to local and regional scale factors. We then computed the afternoon mean residuals from the seasonal cycle and trends using CCGCRV results.*

**S9/ For section 2.4, there is lot of broad qualitative reasoning for instrument issues. I recommend the authors make more extensive use of H16 by referencing the types of QA/QC practise used and provide a quantitative statistical summary of the OPE site, like that for OPE in table 6 of H16. On pg 10, L18 it states, "Raw data is flagged using a set of parameters defined for the station and instrument". This is where a quantitative statistical summary for OPE specifically would be useful.**

We added a table with the quantitative statistical summary of flagging and the following phrases on pages 9:

*For the Picarro G1301 #91, G2301 #379 and G2401 # 728 analysers, the internal flagging parameters are the same as the ones shown on table 4 in Hazan et al. (2016).*

*The list of descriptive flags available to the PI for valid or invalid data is shown on the table 2 of Hazan et al. (2016). The Table 2 presents the quantitative statistical summary of the status of the raw data*

*for the different instruments used at the OPE station. Details of the internal flagging associated with the flags presented in the table below can be found in the table 6 of Hazan et al. (2016). Between 62% and 72% of the raw data are valid while around 25% of the raw data are automatically rejected, 20% being rejected because of stabilisation/flushing.*

**S10/ Table 2 is very complicated and hard to understand. The caption is not helpful. Possibly make a bar plot, as in comment S7, or a table per species.**

The table 2 was simplified and splitted by compound $CO_2$/$CH_4$ and CO as suggested by the reviewer #1

The caption was rephrased as

*Order priority (main vs spare analysers) for the $CO_2$/$CH_4$ compounds with ICOS instrument identifiers and associated period.*

**S11/ Combination of instrument time series. Please detail how priority is set, the instrument with the best precision or 'best' QC/QA? Is there an ICOS procedure to follow for the combination of different instrument timeseries at one location? I find figure 4 a very important piece of information in this manuscript. The current figure does not relay much information. It effectively is central to the OPE total timeseries, as such it would be very helpful to either change figure 4 to display statistics, such as box whisker plots or include another table with the bias and spread of instrument overlap differences. Something akin table 4 in Schibig, et al. (2015). In the final instrument combined time series are the time periods of instrument overlap where there are large differences which instrument measurement is kept? Or should such a disagreement exclude both measurements?**

We added the following phrases on pages 9 and 10 to detail the priority setting :

*From these individual time series, we built three combined time series for $CO_2$, $CH_4$ and CO filling the gaps when possible The objective is to provide users with continuous time series, combining valid measurements in order to minimize the data gaps. Before the merging of the time series each instrument is quality controlled individually, and only measurements which are validated by the automatic data processing and the PI are considered for the combined dataset. For each measurement we indicate the reference of the measuring instrument (unique identifier in the ICOS database), which gives the user the traceability of the analysers taken into account. To build these times series from various analyser datasets we used the priority order given in Table 2 for $CO_2$ and $CH_4$ and Table 3 for CO. The priority order is defined a-priori by the responsible of the station considering which analysers are fully dedicated to the station for long term monitoring purposes. In general secondary instruments are installed for shorter periods to perform specific additional experiments (like dry vs humid air samples, line tests, flushing flow rate tests,etc). For example, 91 was the main instrument for $CO_2$ and $CH_4$ followed by 379. While 91 was in maintenance, instruments 75 or 187 were used as spare instruments. At the beginning of 379 operation, 91 was still the main instrument, to keep the consistency of the time series as long as possible. When 91 operation stopped, 379 becomes the main instrument. When 379 was in repair the instrument 187 was used as spare instrument again. For CO the LGR analyser 80 was the main instrument followed by Picarro G2401 728. When the 80 was out of order, we used either Picarro 187 or LGR 478 as spare instruments. In the case of the installation of two instruments for long term measurements, then the priority order should take into consideration the performance of each one. It is the responsibility of the station manager to change the priority list in the ICOS database if needed.*

Regarding the merging of the individual time series we did not filter out the data with large differences. We did not find any significant time period (days) with systematic large differences. The persistent presence of a bias between two instruments is used as an indication to perform checks on instruments and air intake chains. For important differences, one of the instruments is generally disqualified based on the tests performed. In the case of moderate differences, the objective is to use this information to estimate uncertainties.

We added the following sentences on page 11 line 15:

*No data filtering were applied regarding the differences and the overall biases are small (Table S3). Large differences can be observed on short periods, especially when the atmospheric signal shows very high variability. For such atmospheric conditions any difference in the time lag between air sampling and measurement in the analyser cell has a significant influence. The persistent presence of a bias between two instruments is used as an indication to perform checks on instruments and air intake*

chains. For important differences, one of the instruments is generally disqualified based on the tests performed. In the case of moderate differences, the objective is to use this information for estimating uncertainties.

We added in the supplementary materials a table (S3) showing the statistics (minimum, 1st quartile, median, 3rd quartile, maximum, mean, standard deviations and number of points) of the difference between the afternoon (12:00-17:00 UTC) mean measurements of $CO_2$ and $CH_4$ of the different GHG analysers operated at the same time at the OPE station at the 10m, 50m and 120m levels (figure 5 shows the 120m level plots)

**S12/ There is no mention of the GAW-recommended compatibility limits (GAW, 2011) in section 3 (it is mentioned in the travelling audit section, pg 16, L30). The authors may want to state the GAW compatibility limits and how OPE CMR and LTR compare to these (such in the paragraph starting at pg 12, L15). Does ICOS have a precision and reproducibly limits that needs to be reached? If so this could also be stated and OPE CMR and LTR statistics compare to this guideline instead of the GAW limits.**

**GAW: Report no. 194, 15th WMO/IAEA Meeting of Experts on Carbon Dioxide, Other Greenhouse Gases and Related Tracers Measurement Techniques, Geneva, WMO/TD-No. 1553, 2011.**

ICOS have specific precision and reproducibility limits as shown in the Atmospheric Station specifications report (Laurent, 2017) as well as compatibility goals as WMO/GAW compatibility goals.

| Component | Guaranteed Specification Range | Precision[1] *Std. dev. (1-σ); 1' / 60' average raw data* | Repeatability[2] *Std. dev. (1-σ); 10' average raw data* |
|---|---|---|---|
| $CO_2$ | 350 - 500 ppm | < 50 ppb / 25 ppb | < 50 ppb |
| $CH_4$ | 1700 - 2900 ppb | < 1 ppb / 0.5 ppb | < 0.5 ppb |
| N2O | 300 - 400 ppb | < 0.1 ppb / 0.05 ppb | < 0.1 ppb |
| CO | 30 - 1000 ppb | < 2 ppb / 1 ppb | < 1 ppb |

Test conditions : dry air; room temperature : 20 °C ± 2°C; room pressure: atmospheric pressure with a natural variation.

[1] Measuring a gas cylinder (filled with dry natural air) over 25 hours; first hour rejected (stabilization time).
[2] Measuring alternately a gas cylinder (filled with dry natural air) during 30 minutes and ambient air (not dried) during 270 minutes over 72 hours. Statistics based on the last 10 minute average data of each 30 minute cylinder gas injection (first 20 minutes rejected as stabilization time).

*Table 3 : Gas analyzer performance required by ICOS (as of November 2017)*

We added a phrase regarding ICOS compatibility goal in the indroduction page 2 line 18

In the atmospheric monitoring network, ICOS targets the World Meteorological Organization (WMO) / Global Atmosphere Watch (GAW) compatibility goal (WMO, 2011) within its own network as well as with other international networks.

We added the following phrases in the part 3 : Data Quality Assessment page 12 line 20

As ICOS targets the WMO/GAW compatibility goals within its atmospheric network, the analysers must comply with the performance requirements specified in the Table 3 of the ICOS AS specifications report (Laurent 2017). Precision limits of $CO_2$, $CH_4$ and CO measurements are set to respectively 50 ppb, 1 ppb and 2ppb. Reproducibility limits of $CO_2$, $CH_4$ and CO measurements are set to respectively 50 ppb, 0.5 ppb and 1ppb.

**S13/ CMR monthly means of the time series. Again, a very important part of the manuscript. CMR is related to single instrument performance. Calculating and displaying the CMR of combined instruments does not make sense and contradicts the definition of how the time series is constructed, in the sense that data selection is based upon instrument priority, and exclusion of the lower priority instrument data (pg 9, L2)? A combined CMR in Figure 5 (example 379:187) implies that the timeseries includes all overlapping measurement data. Is this correct? Does ICOS allow this practise? If so, then CMR calculation of a combined dataset should not be performed. I suggest that CMR should be calculated for each individual instrument to be displayed in figure 5. This also applies to LTR statistics in figure 6.**

In the merged minute/hourly time series there are no overlapping data but rather one mixing ratio from one individual analyser for every hour/minute. CMR and LTR are calculated in the ICOS database

for each instrument individually. There is no averaging of uncertainites for multiple instruments performed in the ICOS database. As we show monthly mean results, for some months, several instruments were used and we averaged the individual results. Indeed the meaning of those mixed values are questionable. We have left those values in the figure, where they are clearly labelled with multiple idientifiers, and we have added a warning in the legend.

The merging of individual instruments makes it difficult to display the CMR results for CO on one single plot as Picarro and LGR have very different intrinsic properties. The CO figure has been added in the supplementary material (figure S2).

**S14/ Pg 12, L3, "The time series of CO's CMR o are not shown as the intrinsic properties of the Picarro and Los Gatos Research analysers are very different making it difficult to compare on a same plot.". The performance of the instruments is central to this manuscript; thus, I think it is very important to also present the CO CMRs. CO LTRs are displayed in figure 6. The CO CMRs for the Picarro and Los Gatos can be displayed on separate plots.**

The time series of CO's CMR are now shown in the supplementary material (figure S2)

We changed the phrase by
*The time series of CO's CMR are shown in the supplementary materials (figure S2). The intrinsic properties of the Picarro and Los Gatos Research analysers are very different making it difficult to compare on a same plot .*

**S15/ Table 4. In both Picarro's (187 and 728), LTR is significantly less than CMR. There is no mention of this, or interpretation, as in principle LTR (reproducibility) should be greater than CMR (repeatability). Could the author please comment on this. Pg 15, L11 discusses the Los Gatos instruments but neglects to mention which species they are talking about.**

In fact CMR, as defined in the paper, is closed to the 'precision' value as indicated in analyser datasheet. It is calculated as the standard deviation of the raw data (one point every 1 to 3 sec.) over one minute intervals. LTR indeed is the reproducibility calculated as the standard deviation of target gas injections (averages over several minutes) over 5 days intervals.It is especially true that CMR is larger than LTR for the CO measurements of the G2401 analyser. It means that the raw data display relatively high variabilities, but when averaged over several minutes they are quite stable on few days time scale (meaning the instrument is not very sensitive to temperature/pressure variabilities).

We added the following sentences to the manuscript page14 line 17:

*The Picarro 187 and 728 CO LTR are significantly lower than their CO CMR. This means that their raw data have large high frequency variabilities but when averaged over several minutes these instruments are quite stable (they are not very sensitive to atmospheric or pressure changes).*

Regarding the Los Gatos instruments, the manuscript was modified as below (page 16) :
*While Los Gatos Research instruments show lower CO LTR they have stronger temperature sensitivities generating high short-term variability in conditions where the temperature is not well controlled*

**S16/ Pg 13, L14: "These two types of analysers have very different internal properties making it difficult to show direct comparison.". I disagree with this comment. CMR and LTR can be directly compared and are defined to be independent of instrument internal properties. This is the idea behind using such statistics. Table 4 indicates that instruments 80 and 478 have better CMR and LTR than instruments 187 and 728.**

We agree with the reviewer #1 that CMR and LTR can be compared directly. We changed the phrase by :

*These two types of analysers have very different internal properties as shown on table 5. The CO CMR results reflect such large difference (shown on figure S2), the CO CMR from Los Gatos Research instruments being much lower than the CO CMR from Picarro.*

**S17/ Table 2 shows that the combined times series of CO includes measurements from all four instrument timeseries. This means that the CMR and LTR of the timeseries will have step functions. This should be mentioned in the manuscript (indirectly alluded to at pg 15, L8), preferably referencing H16 (as to how uncertainty estimates are delivered in the end user database).**

We agree with the reviewer #1 that combining different instrument results in steps in the CMR and LTR and in the overall uncertainties

We added the following phrase (line 14 page 10)

*Merging the individual timeseries in such a way implies that the merged time series show steps in their uncertainties as individual analysers have different performance (see part 3 Data Quality Assesment for details about the steps in the repeatability performance).*

**S18/ Pg 15, L25 to pg 16, L11. Just a comment: The audit shows differences. Was there a change in OPE operation due to the audit results?**

The audit shows difference but we were not able to address properly such differences. It was thus quite difficult to change our sampling and measurement strategy without any guess on what to improve. What we learnt from the audit was that we needed a simpler sampling system to use as a spare sampling system and to check for sampling system artefacts.

**S19/ Figure 7, A box whisker plot would convey the target tank statistics a lot clearer with a box whisker plot per tank, per instrument. The cucumber tanks can be left as individual points.**

A box plot would certainly convey the target tank statistics clearly. But the point of this plot is not only to show the statistics but also to show the time behaviour of the bias of the different analysers /sampling system for the three compound. It is important to make sure that there is no major trends, shifts, peaks or steps. It is an important contribution in the overall uncertainty assessment. We thus would like to keep the plot as it is in the initial version. For example, there are periods with large spread of CO bias associated with the temperature sensitivity of the Los Gatos Research analysers. A box plot would not show such period.

*S20/ Pg 18, L7. "A trend may be present". Yes, this is interesting, firstly I thought there was a clear trend, but on reflection there could be a step change at each tank. If the time series is a combination of multiple instrument datasets, then could this be the cause of a possible step change? Would the authors like to comment on possibilities of a continual trend or a series of step changes?*

Due to the high number of instruments there is a relatively high consumption of gases at the OPE station and the lifetime of a target gas is typically limited to 6 months. Figure 8 seems to show an increase of the $CO_2$ concentration measured at the station relatively to the assigned values by the central laboratory. This signal may be due either to a drift in the calibration scale used at OPE, or to step changes in the assigned values of the successive target gases. In order to verify the stability of the calibration scales at longer time scale, ICOS specifications require the use a long-term target gas only after each calibration. The lifetime of this tank is much longer (fifteen to twenty years depending on the instruments number and calibration strategy.

We modified figure 8 to include the long term target results. The long term target $CO_2$ biases also show a slight positive trend (on the order of 0.02 ppm) since 2014 after a step change. Consequently we attribute the signal on figure 8 to the convolution of step changes and possible long term trend. The step changes may be due to cylinders changes. The $CO_2$ biases interannual trend remains unexplained, but all cylinders (calibration and target gases) will be re-evaluated by the ICOS calibration center before the end of their use on site.

We modified the text with the following sentences:

*A slight trend may be present in the LTT $CO_2$ biases between 2014 and 2018. The STT results may show a trend as well but step changes are also present. We attribute the $CO_2$ biases signal to the convolution of step changes and interannual trend. The step changes may be due to cylinders changes. This possible $CO_2$ trend shown by the LTT (on the order of +0.02 ppm) remains unexplained at this stage. The reevaluation of the $CO_2$ concentrations of calibration tanks at ICOS central facility could show a drift in their values, which would lead to a correction of the time series.*

*S21/ Section 4: Results. The first paragraph in this section mentions that general characteristics will be investigated, then diurnal cycles. There also is a need to state that seasonal cycles and long-term trend analysis will also be analysed and commented upon.*

We changed the last part of section 4 by

*We will first show the general characteristics of the time series. We will then analyse and show the diurnal cycles computed from the despiked hourly data. We will select only stable situations with low fast variability to get a focus on the regional scale and compute afternoon stable means for $CO_2$, $CH_4$, CO at the three sampling levels. The seasonal cycles and long-term trend analysis will then be analysed and presented.*

S22/ Section 4.1: General characteristics. Most of this section is about vertical concentration gradients thus should this section be called vertical concentration gradients (or something similar). If this title change is made then 'general characteristics' details can be moved to the appropriate section: diurnal, seasonal or long-term trend. There is also no commentary of the OPE vertical gradients in relation to other tall tower measurements in the same region (or Europe as a whole). Is the drawdown seen at OPE like other measurements? Is it anomalous? This section could use a few more references to contemporary literature to put OPE measurements in context.

We agree that some parts of this section are about vertical gradients. We moved these parts to the section 4.2 which was renamed *diurnal cycles and vertical gradients*. The text in section 4.1 and 4.2 was modified to include references to other European tall tower measurements:

S23/ Figure 9: there are no uncertainty, or spread, bars on these plots. Such uncertainty or spread is critical in such plots and must be displayed.

We added the spread (+ and – 1standard deviations) for each compound and each level on figure 9

S24/ Figure 9: The caption states that the data is normalised to the 120 metre inlet height measurements. Why is this done? I cannot see the reason why. Wouldn't it be better to display the actual non-normalised data? Maybe I am misinterpreting.

We agree that this could be misleading. The diurnal cycle is now presented on figure 9 as actual data (not normalized)

S25/ Figure 9. Are the mean diurnal cycles deseasonalised and detrended? If so (or not) then it should be stated.

The diurnal cycles were not detrended or deseasonalized. We added the following phrase:

*Despiked hourly data (not detrended nor deseasonalized) were used to compute the mean diurnal cycles.*

S26/ Section 4.1 There is no mention of any diurnal cycle in wind direction or speed. Are night time inversions seen? Is the diurnal cycle in CO2, CH4 and CO affected by such inversions or windy nights?

We agree that there is most probably a link between synoptic situations and GHG mole fractions variations (as the last part of the paper suggests). Night time inversions are seen during particular synoptic situations associated with specific wind and trace gases variations. As the paper do not particurlay focus on these aspects, we did not detail such meteorology-GHG relations.

S28/ Section 4.2 should be renamed to something other than the generic title of "data selection and time series analysis", as the section is predominantly concerned with well mixed boundary layer conditions. Data selection is a too generic term. The section should state that data is filtered to represent a well-mixed boundary layer, also state that this filtered data is to be used in seasonal and trend analysis.

We changed the section 4.3 title to *Regional scale signal extraction*

S29/ The 'openair package' and the 'theilsen method' need referencing.

The following references were added:

Sen, P.K.: Estimates of the regression coefficient based on Kendall's tau, Journal of the American Statistical Association, 63 (324): 1379–1389, doi:10.2307/2285891, 1968.

Thoning, K.W., P.P. Tans, and W.D. Komhyr, Atmospheric carbon dioxide at Mauna Loa Observatory, 2. Analysis of the NOAA/GMCC data, 1974 1985., J. Geophys. Res. ,94, 8549 8565,1989

S30/ In the CCGCRV algorithm please specify how was the npoly and nharm variables are set, I.e. using a geophysical basis or iterative attempts to get the best fit?

We used the standard parameters (npoly=3, nharm=4) as our application is quite standard, analysis of afternoon data for 8 years. Pickers and Manning (2015) as well as the man page of CCGCRV recommand the use of these defaults parameters for trend fit with a quadratic function and a four-term harmonic function for the seasonal cycle (in case of seasonal asymetry). The CCGCRV algorithm computes the long terme trend and the seasonal cycle first. Then it filters the residuals to get short

and long term components of the residuals. These were used in the present work and we used the unfiltered residuals.

Pickers, P. A. and Manning, A. C.: Investigating bias in the application of curve fitting programs to atmospheric time series, Atmos. Meas. Tech., 8, 1469-1489, https://doi.org/10.5194/amt-8-1469-2015, 2015.

*S31/ Pg 21, L22. Comparison of CCGCRV residuals with REBS. The sentence on this line states a comparison was made, but no mention of any results of this 'qualitative' comparison. If the comparison was important then results should be mentioned, else maybe leave out the REBs comparison.*

We removed this comparison to keep the paper clearer and simpler

*S32/ Figure 10. Like Fig 9 comments, no 'spread' (1-sigma?) bars for each month. These need to be included. The caption should also state if the seasonal cycles are detrended or not.*

The CCGCRV fitting algorithm does provide uncertainties of the parameters (amplitude and phase of each harmonic functions) but does not assess the overall seasonal cycle uncertainties. As the time series are only 7.5 years long, it is also difficult to compute quantiles for each month or day of the year. We are thus not able to show such spread for the seasonal cycle.

The CCGCRV tool fits a function which approximates the annual cycle and the long term growth in the data. Long term trend estimation are thus deseasonalized and seasonal cycle are detrended. We changed the caption of figure 10 to state it clearly.

*S33/ As in section 4.1, section 4.3 does not mention the seasonal cycle in context of any prior studies. Is the OPE station seasonal cycles anomalous or what is expected. The authors need to put their results into such context.*

The text in section 4.4 was modified to include references to other European tall tower measurements as well as ecosystem flux measurements

*S34/ Pg 23, L18: "We analysed the residuals from the trend...". Residuals from which measurement height? Could the specific height be stated, or all three? (I'm sure it's 120m but should be explicitly stated).*

We changed the phrase to

*« We analysed the 120m level residuals from the trend … »*

*S35/ Table 6. Uncertainty estimates are needed for all calculated trends parameters. Unlike previous sections, the OPE trends are compared to other sites. W. But no mention of the comparisons in respect to OPE or other station trend uncertainties. Please rectify.*

The 95% confidence interval were added for each compound and method in the table 7

*S36/ Figure 11. What is OPE level 3? I gather the 120m height? Maybe remove references to level 3?*

OPE level 3 is the 120m inlet. We removed the reference to level 3 in the figure 11.

*S37/ Pg 25, L23. "We presented the GHG measurement system as well as the quality control performed". Quality control (QC) for OPE was not presented. The QC method used was referenced to H16 and a qualitative description of filtering parameters and issues where given. Explicit OPE filtering diagnostics were not displayed. As stated in S9, the authors already have such statistics available through the ATC processing and should be easily incorporated into the paper.*

Quality control statistics were included as suggested by the reviewer.

*S38/ Section 5 Conclusion: GAW and/or ICOS compatibility limits should be mentioned and referenced when discussing OPE CMR and LTR, travelling standard and target tank results.*

We added the following phrases in the conclusion (page 29, line 17)

*The audits results as well as the routine quality control metrics such as CMR, LTR and biases, and cucumbers intercomparisons showed that the OPE station reached the compatibility goals defined by the WMO for the three compounds, $CO_2$, $CH_4$, and CO for most of the time between 2011 and 2018 (WMO, 2011). The station set-up and its standard operating procedures are also fully compliant with the ICOS specifications (Laurent et al.,. 2017).*

General comments:

The authors presented 8 years of station data, from the Observatoire Perenne de l'Environnement (OPE), which is situated on the eastern edge of the Paris Basin in NE France. As such, this regional station represents continental rural background measurements to the ICOS network and contributes valuable data to link the existing oceanic and urban observation sites. With this study the authors also successfully showed how to interpolate and analyse composite merged data sets, obtained from various sampling analysers in order to comply with stringent ICOS data quality objectives. The paper as a whole is well written and presented and met the objectives set out in the introduction.

The authors would like to thank the anonymous referees #2 for her/his positive general comments.

We changed the manuscript in order to make the improvements suggested. A point-by-point response is included below.

Specific comments :

S1/ Page 10, line10: Prior to this, the authors described differences (in afternoon) between instruments at the same intake height:: this was then followed by a remark that "Schibig et al: : :" found some similar large deviations at their site. Perhaps a better explanation is needed here? Or a table listing the authors' observations in context with other literature reported differences? As it currently reads – it just seemed a bit out of context to me.

This part was modified to include the following sentences

*No data filtering were applied regarding the differences and the overall biases are small (Table S3). Large differences can be observed on short periods, especially when the atmospheric signal shows very high variability. For such atmospheric conditions any difference in the time lag between air sampling and measurement in the analyser cell has a significant influence. The persistent presence of a bias between two instruments is used as an indication to perform checks on instruments and air intake chains. For important differences, one of the instruments is generally disqualified based on the tests performed. In the case of moderate differences, the objective is to use this information for estimating uncertainties.*

*In a similar approach, Schibig et al. (2015) reported results from the comparison between CO2 measurements from two continuous analysers run in parallel at the Jungfraujoch GAW station in Switzerland. The hourly means of the two analysers showed a general good agreement, with mean differences on the order of 0.04 ppm (with a standard deviation of 0.40ppm). However significant deviations of several ppm were also found.*

S2/ Page 12, Lines 15-20: Please put this info in a table format – it makes the intercomparison of the different parameters much easier to read and compare.

A table was included in the supplementary materials (table S4)

S3/ Page17, Figure7: improve y-axis font (make larger); CO bias graph – improve scale to say 2 nmol.mol-1 intervals to show WMO compatibility;

The scale of the plots on figure 8 were improved as suggested by the reviewer.

S4/ Page 23, Lines6-8: I understand the point being made by the authors (i.e. a comparison of observed growth rate at OPE against other nearby sites: : :) but perhaps a better explanation is required when this is compared to Zugspitze? (the Zugspitze growth rate comparison is based on a 1981- 2016 determination: : :) and Cabauw on a 2005 –2009 value for that matter. My question being – Can one draw any useful comparison across such large timescale differences?

We agree with the point made by the reviewer. Such comparison are not quantitative but gives an overview of the published trends recorded at the nearby stations. To make it clear in the paper we added the following sentence:

*Such comparisons are only qualitative and must be used with caution, as the time period considered are different. However, they suggest that the atmospheric $CO_2$ growth may speed up in the European mid-latitudes*

Technical corrections/ comments:

Most of these corrections are as a result of the authors not being English first language speakers and are minor language issues: : :

Page 1, line28: rephrase sentence: : :"Remote and mountain atmospheric measurements: : :"

Page 5, line7: rather use singular for (1) "measurement" and not "measurements"; (2) "ambient air sample" and not "samples"

Page 5, line9: replace "station's " with "stations": : : replace "on" with "in"

Page 6, line8: replace "went first" with either "first went" or "was subjected to: : :"

Page 6, line10: replace "informations" with "information"

Page8, line16: replace "lightnings" with "lightning"

Page 8, line 19: fan, : : :.) add "etc." {et cetera}

Page 9, line9: remove double space after ": : :efficiency)"

Page 10, line4: use plural "sources"

Page11, line29: use singular "measurement"

Page16, Line10: use singular "measurement"

Page18, Line1: replace "to" with "in"

Page 18, Line2: ditto - replace "to" with "in"

Page21, Line30: use plural "dynamics"

Page 21, Line31: add "it" to ": : :seasonal scale make difficult: : :"

Page26, Line10: Rephrase sentence "Interested on larger: : : data"

Page28, Line29: Please check and ensure that the references comply to the journal's requirements "Lowry, D. et al..." Full reference required?

We thank the reviewer for his efforts to improve the manuscript. The previous technical corrections were all taken into account in the revised draft.

---

## Editor Decision (ED1)

Dear Sébastien Conil

Thank you very much for addressing the comments made by the two referees. The revised manuscript improved and should be ready for publication in AMT after addressing the following mainly minor issues listed below.

Best regards,

Christoph Zellweger

**General comments:**

- I recommend the paper be once more thoroughly checked by a native English speaker.
- Large parts of the result section are very descriptive and could be shortened.
- Page1 / Line 16: The precision of the measurements have nothing to do with the compatibility goals, which are a maximal allowed bias. Please revise.
- Figure 5: The differences shown in Figure 5 are relatively large (for hourly averages). Are these hourly averages representing the same data coverage for both instruments, or are the differences mainly due to different temporal coverage within one hour?
- Page 9 / Lines 3-6: Is this needed here? If so, more discussion is needed.
- Page 27, Lines 2-4: 'After a long global decrease since the 1980's, the CO decrease has declined for several years after reaching values below 2 ppm (Lowry et al., 2016, Zellweger et al. 2016).' Something is wrong here. Should it be ppb instead of ppm? Since when has the decrease declined? By how much? (Zellweger et al., 2016) is not an appropriate citation here; (Zellweger et al., 2009) would be better.
- Section 3.3: Scale issues are discussed in this chapter, and it is concluded that the differences can be explained by the bias in the reference scales. If I understood correctly, all comparisons were made on the same calibration scales. Therefore, I would be careful to call this scale issues, since the differences are probably only due to the uncertainties in the mole fraction assignment during calibration, which leads to small biases of the standards.
- Font size in many figures might be too small to be readable in the final AMT paper.
- The term 'concentration' is widely used throughout the manuscript and mixed with other terms such as 'mole fraction' and 'mixing ratio'. Concentration in the context of GHG measurements is not correct and needs to be replaced by mole fraction or amount fraction (mixing ratio might also be acceptable).

**Technical corrections:**

Page1 / Line 16: 'travelling instrument audits' instead of 'travelling instruments audit'

Page1 / Line 18: 'annual growth rates are 2.4 ppm/year and 8.8 ppb/year, respectively, for the' instead of 'annual growth rates are respectively 2.4 ppm/year and 8.8 ppb/year for the'.

Page 1 / Line 19: 'at 120m': be more specific, e.g. 'at 120 m above ground'.

Page 1 / Line 19: 'trend' instead of 'trends'

Page 1 / Line 29: 'For methane' instead of 'As for methane'

Page 2 / Line 6: 'the European' instead of 'European'

Page 2 / Line 19: Please cite the latest GGMT report (WMO, 2018) instead of WMO, 2011

Page 2 / Line 25: Andra and LSCE: Please define acronyms

Page 2 / Line 28: Mace Head is not part of ICOS. The term 'global/mountain station like Mace Head ...' is also confusing. I suggest changing to 'gap between remote stations like ...'.

Page 2 / Line 31: Delete 'France'

Page 4 / Line 4: Please define acronym AS

Page 5 / Lines 6-7: Say also something about cluster 2

Figure 3: I suggest defining FM (flow meter) and PT (pressure transducer)

Page 7 / Line 15: 'Short' instead of 'Shirt'

Page 7 / Line 28: 'one minute' instead of 'minutes'

Page 7 / Line 34: 'defined' here seems awkward. Consider to re-phrase, e.g. 'All standards were calibrated following ....'

Page 8 / Line 3: Wrong scale for CH4, please correct (should be probably WMO-2004)

Page 8 / Line 5: Is the CH4 scale here correct (WMO-2004), or should it be WMO-2004A? Scale for N2O also seems to be incorrect; it should be either WMO-2006 or WMO-2006A. Scale for CO: should it be WMO-2014A (instead of WMO-X2014). Please also use consistent scale nomenclature (currently with and without X etc.)

Page 8 / Line 6: 'measurement' instead of 'the measurements'

Page 8 / Line 15: 'information' instead of 'informations', also instrumentation

Table 1: Define N and K (reference to Hazan et al. is not sufficient). Caption: 2018 2014 – please correct.

Page 13 / Line 6: Please define acronym MLab

Page 13 / Line 16: G2401 instead of G2400

Page 13 / Line 19: 'this information' instead of 'these informations', Table instead of Ttable

Page 15 / Line 1: Title (3.2) should be revised. I cannot understand what is meant here.

Page 17 / Line 23: 'dried Picarro' and 'wet Picarro', and similar expressions later: colloquial, please rephrase.

Page 18 / Line 2: The sentence 'deviations were either higher than or barely within the ...' should be revised. 'Higher than' and 'barely within' both mean that they were exceeding the goal.

Page 18 / Line 15: Replace 'The aims of such programs are' with 'The aim of these programs is'

Figure caption, Fig. 8: Delete 'in colours' and 'respectively'.

Page 21 / Line 6: Would Zugspitze-Schneefernerhaus (ZSF) be a better choice compared to ZUG? The global GAW site is now ZSF, not ZUG.

Figure 10: There are three lines (red, green, blue) and also the corresponding shaded areas in the same colours, but there is an additional shaded area which is not mentioned. What is this?

Page 23, Line 27/28: Rephrase. Suggestion: 'The station time series exhibit strong variability from hourly to interannual time scales. These variations may be related to meteorological variability, and to variations in the sources and sinks.'

Page 24, Line 6. Full stop after efficient, not after more.

Page 24, Line 33, and Page25, Line 5: To my knowledge, Yuan et al. do not discuss JFJ data in their paper. Please add a reference for the JFJ seasonal cycle.

Page 29, Line 7: Suggest rephrasing to 'sampling regionally representative air-masses'

Page 29, Line 10: ERIC not defined.

The above list is not complete. There is still potential to further improve the manuscript as suggested by reviewer #1. Another check on language issues, and another attempt to further condense and optimize the structure of the paper should made.

**References:**

WMO: 19th WMO/IAEA Meeting on Carbon Dioxide, Other Greenhouse Gases and Related Tracers Measurement Techniques (GGMT-2017), Dübendorf, Switzerland, 27-31 August 2017, GAW Report No. 242, World Meteorological Organization, Geneva, Switzerland, 2018.

Zellweger, C., Emmenegger, L., Firdaus, M., Hatakka, J., Heimann, M., Kozlova, E., Spain, T. G., Steinbacher, M., van der Schoot, M. V., and Buchmann, B.: Assessment of recent advances in measurement techniques for atmospheric carbon dioxide and methane observations, Atmos. Meas. Tech., 9, 4737-4757, 2016.

Zellweger, C., Hüglin, C., Klausen, J., Steinbacher, M., Vollmer, M., and Buchmann, B.: Inter-comparison of four different carbon monoxide measurement techniques and evaluation of the long-term carbon monoxide time series of Jungfraujoch, Atmos. Chem. Phys., 9, 3491-3503, 2009.

---

## Author Response (AR2)

**Reply to the Editor**

We would like to thank the editor for his efforts to improve the manuscript. We address his general comments below. In the following, editor's comments are given in bold, author's responses in plain text. Suggested new text is quoted in italics.

**General Comments**

**G1/ I recommend the paper be once more thoroughly checked by a native English speaker**

The manuscript has now been corrected by a professional translation and editing company

**G2/ Large parts of the result section are very descriptive and could be shortened.***

In compliance with this comment, we have shorterted the results section as much as we could.

**G3/ Page1 / Line 16: The precision of the measurements have nothing to do with the compatibility goals, which are a maximal allowed bias. Please revise.**

We have modified the text as suggested by the editor:

*Thanks to the quality assurance strategy recommended by ICOS, the measurements uncertainties are within the World Meteorological Organisation compatibility goals for carbon dioxide ($CO_2$), methane ($CH_4$) and carbon monoxide (CO).*

**G4/ Figure 5: The differences shown in Figure 5 are relatively large (for hourly averages). Are these hourly averages representing the same data coverage for both instruments, or are the differences mainly due to different temporal coverage within one hour?**

We agree with the editor that the differences are relatively large for hourly average. As stated in the manuscript, the "significant deviations may come from various sources of uncertainty, such as different residence time in the sampling systems, water vapour correction, clock issues, or internal analyser uncertainties." The differences may also come from differences in the sequence of air sampling, as we use a multi level air sampling system. The hourly means are computed based on only 15 minutes from the hour. The sampling sequencers were synchronized but sometimes the synchronization drifted or was lost for various reasons. If the sequencers are not synchronized, the air samples sampled are not the same and that can introduce large differences. We focused on the afternoon to reduce short term variabilities and to reduce local effects. Nevertheless, there are afternoon situations were short term variabilities are important and they were not filtered out.

We modified the following sentence to clarify this point:

*For such atmospheric conditions, any difference in the air sampling sequences or in the time lag between air sampling and measurement in the analyser cell has a significant influence*

**G5/ Page 9 / Lines 3-6: Is this needed here? If so, more discussion is needed**

We believe that this is useful information. Ancillary data are used to perform the automatic flagging of the raw data from the analysers. We modified the text as below:

*Sequence data are used to generate the ambient air and cylinders raw time series. Mole fractions raw data are flagged automatically using the ancillary data based on a set of parameters defined for each station and instrument.*

**G6/ Page 27, Lines 2-4: 'After a long global decrease since the 1980's, the CO decrease has declined for several years after reaching values below 2 ppm (Lowry et al., 2016, Zellweger et al. 2016).' Something is wrong here. Should it be ppb instead of ppm? Since when has the decrease declined? By how much? (Zellweger et al., 2016) is not an appropriate citation here; (Zellweger et al., 2009) would be better..**

We agree with this comment. We corrected the sentence and citations to

*After a global decrease since the end of the 1980s, CO decrease has declined for several years after reaching values below 0.2 ppm at European background sites MHD or JFJ (Lowry et al., 2016, Novelli et al., 2003; Zellweger et al., 2009).*

G7/Section 3.3: Scale issues are discussed in this chapter, and it is concluded that the differences can be explained by the bias in the reference scales. If I understood correctly, all comparisons were made on the same calibration scales. Therefore, I would be careful to call this scale issues, since the differences are probably only due to the uncertainties in the mole fraction assignment during calibration, which leads to small biases of the standards.

Indeed, there is no calibration scale issue as both systems are using the same primary WMO scale ($CO_2$: WMO X2007 $CH_4$: WMO X2004A CO: WMO X2014A). The observed bias is related to the uncertainty on mole fraction assignment of the 2 different sets of working standards used independently by the station onsite analysers and the FMI's travelling instruments. These 2 sets of standards have been calibrated against the same WMO scale but using 2 different tertiary standards (FMI and LSCE) with 2 independent setups and associated data processing. The multiplicity of intermediary standards in the calibration chain increases uncertainties and may lead to bias in the 2 different sets of calibrations standards.

A misuse of "scale" instead of "calibration standards" in the text leads to this confusion.

We modified the following sentence to clarify this point:

*Most of the differences in ambient air measurements can be explained by biases in the working standards. The instruments and the working standards (OPE and travelling standards) were calibrated against two different sets of tertiary standards, introducing biases in the measurements of the cylinders and of ambient air.*

G8/ Font size in many figures might be too small to be readable in the final AMT paper.

We could make new plots with larger font size if needed

G9/ The term 'concentration' is widely used throughout the manuscript and mixed with other terms such as 'mole fraction' and 'mixing ratio'. Concentration in the context of GHG measurements is not correct and needs to be replaced by mole fraction or amount fraction (mixing ratio might also be acceptable).

We replaced concentration by mole fraction throughout the manuscript

Technical corrections:

T1: done

T2: done

T3: done

T4: done

T5: done

T6: done

T7: done

T8: done

T9: done

T10: done

T11: done

T12: We modified the sentence to: *Clusters 1, 2 and 3 are characterised by continental air masses (mostly from the south, east and north respectively)*

T13: done

T14: done

T15: done

**T16:** We modified the sentence to: *The standards mole fractions cover the unpolluted atmospheric range following ICOS Atmospheric Station specifications (Laurent, 2017).*

**T17:** done

**T18:** the scale references were corrected

**T19:** done

**T20:** done

**T21:** done

**T22:** done

**T23:** done

**T24:** done

**T25:** Title 3.2 was simplified to *Field long term repeatability*

**T26:** 'dried Picarro' and 'wet Picarro', and similar expressions were removed

**T27:** The sentence was changed to *The CO comparison was carried out for OPE-LGR and OPE-G2401 instruments and compared to the TI G2401: the average deviations exceeded the WMO/GAW component compatibility goal (±2 ppb).*

**T28:** done

**T29:** done

**T30:** done

**T31:** There are only 3 colours in the plots, the other colour being a mixture of red, green and blue for the part of the plots were the areas are common to the 3 colours

**T32:** The text was modified as suggested

**T33:** done

**T34:** Reference to JFJ was removed

**T35:** done

**T36:** ERIC (European Research Infrastructure Consortium) was removed from the sentence

[revised manuscript text omitted]

---

## Author Response (AR3)

**Reply to the Editor**

We would like to thank the editor for his last efforts to improve the manuscript. We corrected the manuscript following his suggestions. We have also carefully checked the manuscript and corrected several other typos.

We also modified the manuscript regarding the following issues:

- Page 8, lines 4-8: the scales were checked and the sentences were modified accordingly

- Page 26, line 13: the sentence was simplified as

  *This finding is consistent with recent observations in Europe and in the USA (Lowry et al., 2016, Novelli et al., 2003; Zellweger et al., 2009).*

- Page 28, line 27: this part of the conclusion was changed according to the editor's suggestion

- Data availability and author contributions sections were added

[revised manuscript text omitted]